# Development of a global 30-m impervious surface map using multi-source and multi-temporal remote sensing datasets with the Google Earth Engine platform

Xiao Zhang[1, 2], Liangyun Liu[1, 2], Changshan Wu[3], Xidong Chen[1, 2], Yuan Gao[1, 4], Shuai Xie[1, 2] and Bing Zhang[1, 2]

[1] State Key Laboratory of Remote Sensing Science, Aerospace Information Research Institute, Chinese Academy of Sciences, Beijing 100094, China
[2] University of Chinese Academy of Sciences, Beijing 100049, China
[3] Department of Geography, University of Wisconsin-Milwaukee, Milwaukee, WI, USA
[4] College of Geomatics, Xi'an University of Science and Technology, Xi'an 710054, China

*Correspondence to*: Liangyun Liu (liuly@radi.ac.cn)

**Abstract.** The amount of impervious surface is an important indicator in the monitoring of the intensity of human activity and environmental change. The use of remote sensing techniques is the only means of accurately carrying out global mapping of impervious surfaces covering large areas. Optical imagery can capture surface reflectance characteristics, while synthetic aperture radar (SAR) images can be used to provide information on the structure and dielectric properties of surface materials. In addition, night-time light (NTL) imagery can detect the intensity of human activity and thus provide important a priori probabilities of the occurrence of impervious surfaces. In this study, we aimed to generate an accurate global impervious surface map at a resolution of 30-m for 2015 by combining Landsat-8 OLI optical images, Sentinel-1 SAR images and Visible Infrared Imaging Radiometer (VIIRS) NTL images based on the Google Earth Engine (GEE) platform. First, the global impervious and non-impervious training samples were automatically derived by combining the GlobeLand30 land-cover product with VIIRS NTL and MODIS enhanced vegetation index (EVI) imagery. Then, the local adaptive random forest classifiers, allowing a regional adjustment of the classification parameters to take into account the regional characteristics, were trained and used to generate regional impervious surface maps for each $5\degree \times 5\degree$ geographical grid using local training samples and multi-source and multi-temporal imagery. Finally, a global impervious surface map, produced by mosaicking numerous $5\degree \times 5\degree$ regional maps, was validated by interpretation samples and then compared with five existing impervious products (GlobeLand30, FROM_GLC, NUACI, HBASE and GHSL). The results indicated that the global impervious surface map produced using the proposed multi-source, multi-temporal random forest classification (MSMT_RF) method was the most accurate of the maps, having an overall accuracy of 95.1% and kappa coefficient (one of the most commonly used statistics to test interrater reliability (Olofsson et al., 2014) ) of 0.898 as against 85.6% and 0.695 for NUACI, 89.6% and 0.780 for FROM_GLC, 90.3% and 0.794 for GHSL, 88.4% and 0.753 for GlobeLand30, and 88.0% and 0.745 for HBASE using all 15 regional validation data. Therefore, it is concluded that a global 30-m impervious surface map can accurately and efficiently

be generated by the proposed MSMT_RF method based on the GEE platform. The global impervious surface map generated in this paper is available at https://doi.org/10.5281/zenodo.3505079 (Zhang and Liu, 2019).

## 1 Introduction

Impervious surfaces are usually covered by anthropogenic materials which prevent water penetrating into the soil (Weng, 2012), which are primarily composited by asphalts, sand and stone, concrete, bricks, glasses, etc. (Chen et al., 2015). Due to the rapid growth in the area covered by impervious surfaces, a series of climate, environmental and social problems are emerging, including the urban heat island, traffic congestion, waterlogging and the deterioration of urban environment (Fu and Weng, 2016; Gao et al., 2012; Weng, 2001; Zhou et al., 2017; Zhuo et al., 2018). Furthermore, as an important indicator in the

monitoring of the intensity of human activity and of ecological and environmental changes, the mapping of impervious surfaces is of great interest in many disciplines (Xie and Weng, 2017). Accurate large-area impervious surface mapping is, therefore, urgent and necessary.

Due to the frequent and large-area coverage that it provides, increasing attention has been paid to the use of remote sensing technology for impervious surface mapping. In recent years, a lot of effort has gone into mapping impervious surfaces at

different spatial resolutions (Elvidge et al., 2007; Schneider et al., 2010; Schneider et al., 2009). For example, Schneider et al. (2010) used multi-temporal MODIS data to produce a 500-m global urban land map, achieving an overall accuracy of 93% and kappa coefficient of 0.65. Elvidge et al. (2007) combined the Defense Meteorological Satellite Program (DMSP) Operational Linescan System (OLS) and LandScan population count data to produce a 1-km global impervious surface area map. However, since the complex characteristics of impervious landscapes and inherent resolution of human activity, coarse-

resolution global impervious surface maps were not suitable for many applications and policy makers at local or regional scales, for example, the urban-rural pattern planning and road network monitoring usually required the fine spatial resolution impervious surface products (Gao et al., 2012).

Recently, with the advent of free medium-resolution satellite data (e.g. Landsat and Sentinel-2), combined with rapidly-increasing data-storage and computation capabilities, many regional or global fine-resolution impervious surface maps have

been produced using Landsat and Sentinel-2 images (Chen et al., 2015; Gao et al., 2012; Goldblatt et al., 2018; Gong et al., 2019; Gong et al., 2013; Homer et al., 2015; Li et al., 2018; Liu et al., 2018; Sun et al., 2017). Specifically, the National Land Cover Dataset (NLCD) produced the first 30-m map of the United States including impervious surface as three separate land-cover types (Developed low, Developed medium, and Developed high intensity) using Landsat imagery, DMSP OLS and United States Geological Survey (USGS) National Elevation Dataset (NED) digital elevation data, and achieving the user'

accuracy of 0.48~0.66 (Homer et al., 2004). Similarly, the Finer Resolution Observation and Monitoring of Global Land Cover (FROM_GLC) produced the global 30-m impervious surface map as an independent land cover type with the user' accuracy of 0.307 (Gong et al., 2013); the 30 m Global Land Cover data product (GlobeLand30) combined the pixel-based classification,

segmentation and manual editing based on the high resolution imagery to develop the 30-m impervious surface map as an independent layer with the user's accuracy of 0.867 (Chen et al., 2015). However, as sparse training samples of impervious surfaces cannot capture all relevant spectral heterogeneity when producing these land-cover products, the impervious surface layers usually suffered low accuracy except for GlobeLand30 (which includes manual interpretation). Therefore, a few studies proposed to independently produce the impervious surface products. For example, Liu et al. (2018) proposed the Normalized Urban Areas Composite Index (NUACI) method to produce a global 30-m impervious surface map and achieved an overall accuracy of 0.81-0.84 and a kappa values of 0.43-0.50. However, the NUACI product had a relatively poor performance in terms of producer's accuracy (0.50–0.60) and user's accuracy (0.49-0.61). Brown de Colstoun et al. (2017) combined the object-based segmentation, random forest classification and post-processing to develop the Global 30-m Man-made Impervious Surface (GMIS) and Human Built-up and Settlement Extent (HBASE) dataset in 2010 which achieved a kappa coefficient of 0.91 using scene-level cross validation in Europe (Wang et al., 2017b). Pesaresi et al. (2016) used the multi-temporal Landsat imagery and symbolic machine learning method to produce the Global 30-m Human Settlement Layer (GHSL) in 2014, and achieved a total accuracy of 96.28% and kappa coefficient of 0.3233 based on Land Use/Cover Area frame Survey (LUCAS) reference data. Therefore, an accurate impervious surface map at fine spatial resolution is still urgently needed using an efficient mapping method.

There are three critical challenges for global impervious surface mapping at medium spatial resolution. These include finding an adequate image identification method, image selection scheme and image processing platform (Liu et al., 2018).

First, although a wide range of methods have already been presented for impervious surface mapping, it is still hard to generate an operational and accurate global impervious surface map at 30-m resolution. The methods used so far can be divided into three main groups: spectral mixture analysis methods (Ridd, 1995; Wetherley et al., 2017; Wu, 2004; Wu and Murray, 2003; Yang and He, 2017; Zhuo et al., 2018), spectral index-based methods (Deng and Wu, 2012; Liu et al., 2018; Xu, 2010), and image classification methods (Chen et al., 2015; Okujeni et al., 2013; Zhang et al., 2018a; Zhang et al., 2012; Zhang and Weng, 2016). The spectral mixture analysis methods have great advantages in terms of the repeatable and accurate extraction of quantitative sub-pixel information (Weng, 2012). However, these spectral mixture methods can produce underestimates in areas with high density impervious surfaces and overestimates in areas with low density impervious surfaces, and may have great difficulties to identify one suitable endmember to represent all types of impervious surfaces (Sun et al., 2017; Weng, 2012). The spectral index-based methods have been widely applied in regional impervious surface mapping due to their simplicity, flexibility and convenience (Liu et al., 2018; Sun et al., 2019b; Xu, 2010). However, the spectral index-based methods have great difficulty in finding the optimal threshold for separating the impervious pixels from bare areas and vegetation pixels (Sun et al., 2017). The image classification methods can efficiently combine remote sensing datasets from multiple sources (Zhang et al., 2016; Zhang et al., 2018a; Zhou et al., 2017) and have great capabilities in spectrally complex impervious surface mapping (Okujeni et al., 2013), which has been an area of great interest in recent years (Goldblatt et al.,

2018; Zhang et al., 2018b). However, it is very hard to select training samples for large-area impervious surface mapping using these methods (Weng, 2012).

Second, although individual optical data sets have been successfully employed for regional or global impervious surface mapping, accurate estimation of impervious surfaces remains challenging due to the diversity of urban land-cover types, which leads to difficulties in separating different land-cover types with similar spectral signatures (Zhang et al., 2014b). The

incorporation of multi-source and multi-temporal remote sensing imagery has been demonstrated to improve impervious surface mapping accuracy (Weng, 2012; Zhu et al., 2012). For example, optical imagery is only able to capture surface reflectance characteristics, while synthetic aperture radar (SAR) data can provide details of the structure and dielectric properties of the surface materials (Sun et al., 2019b; Zhang et al., 2014b; Zhu et al., 2012). Zhang et al. (2016) found that the addition of dual-polarimetric SAR features resulted in an accuracy improvement of 3.5% compared with using optical SPOT-

5 imagery only and also that dual-polarimetric SAR data had a superior performance to single polarimetric SAR data for impervious mapping. Similarly, Shao et al. (2016) explained that the combination of GaoFen-1 optical imagery with Sentinel-1 SAR imagery efficiently reduced the confusion between impervious surfaces and water and bare areas. Furthermore, Zhu et al. (2012) found that the inclusion of multi-seasonal imagery increased the mapping accuracy from 77.96% to 86.86% and that the further addition of texture variables increased the mapping accuracy to 92.69% for urban and peri-urban land-cover

classification. The reasons for the accuracy increase were that the texture imagery could capture the local spatial structure and the variability in land cover categories and also that the temporal information could describe the phenological variability. Schug et al. (2018) also used the multi-seasonal Landsat imagery to successfully map impervious extent and land cover fractions. In addition, as an important data source for the measurement of socioeconomic activities, DMSP-OLS night-time light (NTL) imagery have been widely used in many impervious-related applications (Li and Zhou, 2017). For example,

Elvidge et al. (2007) successfully produced a global 1-km impervious map using DMSP-OLS NTL imagery, Goldblatt et al. (2018) combined DMSP-OLS NTL and Landsat-8 imagery to accurately produce 30-m impervious surface maps at a national scale. Therefore, the integration of multi-source and multi-temporal datasets is necessary and crucial to the production of accurate global impervious surface maps.

Lastly, the mapping of impervious surfaces at the global scale usually requires huge amounts of computation and large storage

capabilities. Fortunately, the Google Earth Engine (GEE) cloud-based platform consists of a multi-petabyte analysis-ready data catalog co-located with a high-performance, intrinsically parallel computation service (Gorelick et al., 2017), meaning that the requirements for large-area image collection and very large computational resources can easily be met by using the free-access GEE cloud-computation platform. For example, Liu et al. (2018) produced multi-temporal global impervious surface maps and Pekel et al. (2016) developed global high-resolution surface water maps and analyzed long-term changes

using the GEE cloud-computation platform. Recently, Massey et al. (2018) produced a continental-scale cropland extent map for North America at 30 m spatial resolution for the nominal year 2010 based on the GEE platform. It can be seen, therefore, that the GEE is an efficient and useful computation platform for regional/global applications.

So far, due to the limitations of data collection and computation capability, impervious surface mapping has mainly focused on using a single type of remote sensing data or on case studies made at the regional scale. Although the GEE platform provides multi-petabyte analysis-ready data and efficient data-processing capabilities, an efficient method that can fully integrate these multi-source and multi-temporal datasets and produce accurate impervious surface maps at a spatial resolution of 30-m for the whole world is still lacking. The aims of this study, therefore, were (1) to produce a global 30-m impervious surface map from multi-source multi-temporal remote sensing datasets including Landsat-8 OLI, Sentinel-1 SAR, VIIRS NTL and MODIS imagery using the GEE platform; (2) to investigate the accuracy of the global 30-m impervious surface mapping using validation samples and cross-comparison with five existing impervious surface products (GlobeLand30 (Chen et al., 2015), FROM_GLC (Gong et al., 2013), NUACI (Liu et al., 2018), GHSL (Florczyk et al., 2019) and HBASE (Wang et al., 2017a)). The results indicate that the global impervious surface map produced by the proposed method is accurate and is suitable for regional or global impervious surface applications.

## 2 Datasets

### 2.1 Remote sensing datasets

In this study, three kinds of data sources including Landsat-8 optical imagery, Sentinel-1 SAR data and digital elevation model (DEM) topographical variables were selected and collected for the mapping of impervious surfaces across the world on the GEE platform. Furthermore, the combination of VIIRS NTL imagery and MODIS Enhanced vegetation index (EVI) products was used to derive the set of global impervious surface and non-impervious surfaces training data.

All available Landsat-8 surface reflectance (SR) imagery from 2015 and 2016 (USGS, 2015), which had been archived on the GEE platform, were used in this study for the nominal year 2015 because of the frequent cloud contamination in the tropic areas. All the SR images were radiometrically corrected by the Landsat Surface Reflectance Code (LaSRC) atmospheric correction method (Hu et al., 2014; Vermote et al., 2016), and bad pixels including clouds, cloud shadow, and saturated pixels were identified by the CFMask algorithm (Guide, 2018).

The Sentinel-1 satellite provides C-band SAR imagery at a variety of polarizations and resolutions (Berger et al., 2012; ESA, 2016; Torres et al., 2012). Due to the high dielectric properties of the building materials, the unique geometry of manmade features, and the special radar echo properties of artificial structures, the impervious surfaces usually had stronger backscattered signals than other land-cover types (such as: barren land, cropland and so on) in the SAR imagery. In this study, all available Sentinel-1 imagery from 2015 and 2016, which had already been calibrated and ortho-corrected then archived on the GEE platform, were also used for the nominal year 2015. In addition, each Sentinel-1 image on the GEE had been pre-processed with the Sentinel-1 Toolbox, including thermal noise removal, radiometric calibration and terrain correction (https://developers.google.com/earth-engine/sentinel1). Also, as HH- and HV-polarized Sentinel-1 SAR imagery does not cover the whole world (Sun et al., 2019a), a combination of dual-band cross-polarized (VV and VH) Interferometric Wide

Swath (IW) mode imagery in both 'ascending' and 'descending' orbits was used. The spatial resolution of this imagery was 10-m and the repeat cycle of the polar-orbiting two-satellite constellation is 6 days.

The Shuttle Radar Topography Mission (SRTM) DEM, provided by the NASA JPL at a resolution of 1 arc-second (approximately 30 m) and covering the area between 60 ° north and 56 ° south (Farr et al., 2007), was used as an auxiliary dataset for impervious surface mapping, because numerous studies have demonstrated that the spatial distribution of impervious surfaces was related to the topographical variables (Ban et al., 2015; Sun et al., 2019b). For example, Sun et al. (2019b) used a slope threshold to exclude impervious surface over mountain areas if the slope was larger than 15˚ for impervious mapping in China. This dataset has undergone a void-filling process using other open-source data (ASTER GDEM2, GMTED2010 and NED) in the GEE platform. As for the high-latitude areas that lacked the SRTM data, the Advanced Spaceborne Thermal Emission and Reflection Radiometer (ASTER) Global Digital Elevation Model Version 2 (GDEM V2) (Tachikawa et al., 2011) was used instead.

The VIIRS NTL, collected by NASA/NOAA's Suomi National Polar-orbiting Partnership satellite (https://maps.ngdc.noaa.gov/viewers/VIIRS_DNB_nighttime_imagery/index.html), has the unique ability to record emitted visible and near-infrared (VNIR) radiation at night with a spatial resolution of 15 arc seconds (equivalent to 0.5 km at the equator) (Elvidge et al., 2017). Compared to the DMSP-OLS NTL data, the VIIRS NTL data provide higher spatial resolution, and finer radiometric resolution, which allows weaker surface radiation to be detected (Bennett and Smith, 2017). It is also the main data source used for studying the expansion of impervious surfaces and related sociodemographic issues (Elvidge et al., 2017). In this study, a combination of VIIRS NTL, MODIS EVI imagery and GlobeLand30 land-cover products was used to derive the set of global training samples.

The MODIS EVI imagery (MYD13Q1) from the MODIS V6 products contains the best available EVI data from among all the acquisitions obtained over a 16-day compositing period and has a spatial resolution of 250-m (Didan et al., 2015), which was used to mitigate the NTL data's saturation problem and exclude false positive impervious samples (vegetated samples in the urban) when deriving the global training samples. In this study, the EVI imagery for 2015 in the GEE used the blue band to remove residual atmospheric contamination caused by smoke and sub-pixel thin clouds (https://developers.google.com/earth-engine/datasets/catalog/ MODIS_006_MYD13Q1).

## 2.2 Global impervious surface products

In this study, five global impervious surface products (GlobeLand30, FROM_GLC, NUACI, HBASE and GHSL) − were used to validate the global impervious surface map produced using the multi-source, multi-temporal random forest classification (MSMT_RF) method. The GlobeLand30 data were also used to automatically derive the global impervious and non-impervious training samples.

GlobeLand30 is an operational 30-m global land-cover dataset produced using the Pixel-Object-Knowledge-based method (POK-based) approach in 2000 and 2010 (Chen et al., 2015). In this study, the global impervious product derived from GlobeLand30 in 2010 (GlobeLand30-2010, http://www.globallandcover.com/GLC30Download/index.aspx) was produced by combining pixel-based classification, multi-scale segmentation and manual editing based on the high resolution imagery and had been validated as having a user's accuracy of 86.7%.

FROM_GLC, first produced in 2010, was the first 30-m resolution global land-cover dataset and was produced by supervised classification of 8,900 Landsat images (Gong et al., 2013). In this study, the second generation of FROM_GLC from 2015 (FROM_GLC-2015) (http://data.ess.tsinghua.edu.cn/) was used. This dataset was produced by using multi-seasonal Landsat imagery acquired between 2013 and 2015 and incorporates the day of year, geographical coordinates and elevation data (Li et al., 2017).

The NUACI-based maps, developed using the spectral index-based method applied to Landsat and DMSP-OLS NTL imagery, are multi-temporal global 30-m impervious surface datasets (Liu et al., 2018). In this study, the NUACI impervious map from 2015 (NUACI-2015) was used (http://www.geosimulation.cn/ GlobalUrbanLand.html). This map has been validated as having an overall accuracy of 0.81–0.84 and kappa coefficient of 0.43–0.50 at the global level (Liu et al., 2018).

The HBASE dataset was the first global 30-m dataset of man-made impervious cover derived from the Global Land Survey (GLS) Landsat data for 2010 (HBASE-2010) (https://sedac.ciesin.columbia.edu/data/set/ulandsat-hbase-v1). It was produced by combining meter-resolution training data (exceeding 20 millions), Open Street Map, VIIRS NTL, GLS Landsat SR and MODIS NDVI products, and achieved a kappa coefficient of 0.91 using scene-level cross validation in Europe (Wang et al., 2017a; Wang et al., 2017b).

The GHSL, a global information baseline describing the spatial evolution of the human settlements in the past 40 years, was developed by using symbolic machine learning model trained by the collected high-resolution samples, multi-temporal Landsat imagery in the epochs 1975, 1990, 2000, and 2015 (Florczyk et al., 2019). In this study, the GHSL impervious surface map at 30-m for 2015 (GHSL-2015) (https://ghsl.jrc.ec.europa.eu/download.php) was employed for comparison analysis, which achieved an overall accuracy of 96.28% and kappa coefficient of 0.3233 validated using Land Use/Cover Area frame Survey reference data (Pesaresi et al., 2016).

### 2.3 Validation samples

To quantitatively assess the performance of the global impervious surface datasets, fifteen validation regions, covering different continents and various urban landscapes (the bare soil prevalent cities: Phoenix (PNX), Madrid (MDR), Riyadh (RYH), Niamey (NIM), Johannesburg (JHB), Ntuman (NTU) and Lhasa (LHS), vegetation prevalent cities: New York (NYK), Manaus (MNS), Moscow (MSC), San Paulo (SPL) and Melbourne (MBN), as well as cropland prevalent cities: Winnipeg (WIP), Bangkok (BGK) and Xi'an (XAN)), were selected (Fig. 1). For each validation region, 600-1000 samples were

randomly generated using the stratified random sampling strategy (Bai et al., 2015). As there were significant advantages to using Google Earth for validation sample selection (Zhang et al., 2018c), each sample was labeled either as "non-impervious surface" or "impervious surface" based on visual interpretation of the available high-resolution remote sensing imagery in Google Earth. To ensure the reliability of each validation sample, two prior impervious products, including NLCD impervious products (Homer et al., 2015) and Copernicus land monitoring surface – high resolution layer imperviousness (Langanke et al., 2016) which were validated to achieve high overall, user's and product's accuracies exceeding 82% and 90% respectively, were overlaid to the high-resolution remote sensing imagery. In addition, the location of each sample was moved to the center of the relevant surface object (building, road, etc.) because of the greater spectral mixing effect and uncertainty at the boundary of the objects. Like the work of Sun et al. (2019b), if the impervious area in the 30-m ×30-m validation window was more than a predefined threshold of 50%, we will consider this validation point as impervious surface, otherwise, it would be labeled as non-impervious surface. After careful interpretation, a total of 11,942 samples including 4952 impervious samples and 6990 non-impervious samples were obtained. In order to minimize the subjective influence of interpretation, the validation samples were collected independently by three different scientists. If there was dispute between the interpretation results of three scientists, the validation point was discarded.

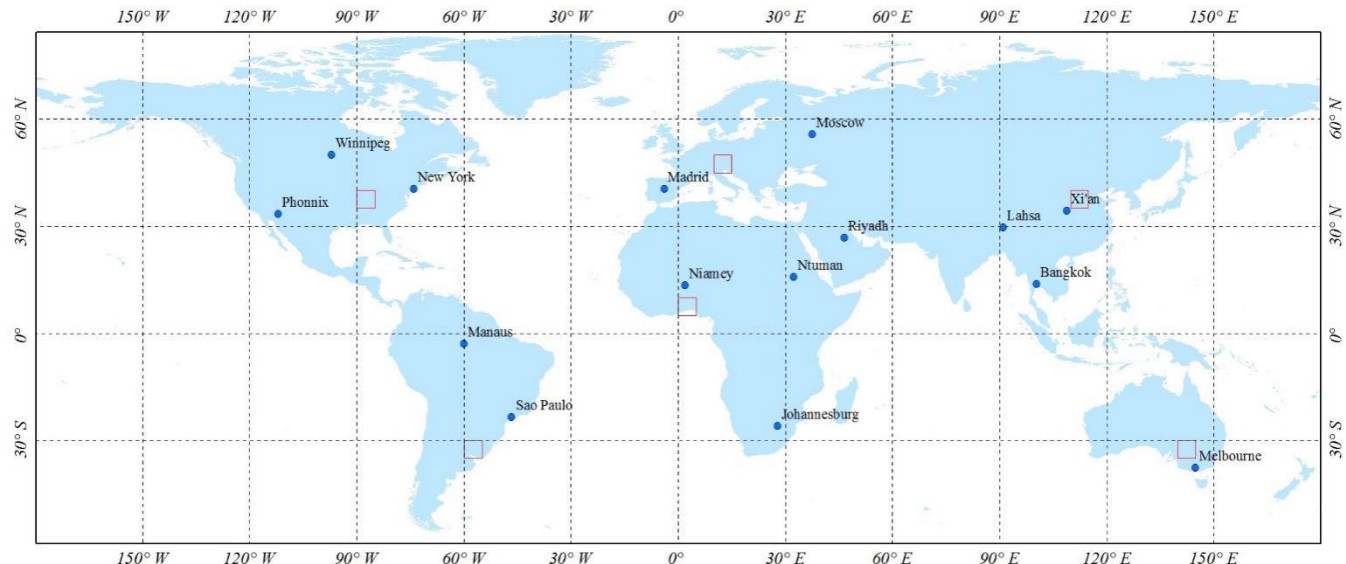

**Figure 1: The spatial distribution of the fifteen validation regions (blue) corresponding to regions of different impervious landscapes on different continents together with the six 5 °×5 °validation regions (red) used to measure the variable importance.**

### 3 Collection of global training samples

As the reliability and representativeness of the training samples would affect the classification accuracy directly (Foody and Mathur, 2004), we proposed combining GlobeLand30, VIIRS NTL and MODIS EVI data to derive accurate impervious and non-impervious samples. The GlobeLand30 land-cover product was used to derive global training samples because it had

many advantages including: (1) the impervious surface layer in GlobeLand30 was accurately developed by combining the pixel-based classification, multi-scale segmentation and manual editing based on high resolution imagery and validated to achieve an user's accuracy of 86.7%; (2) it simultaneously contained the impervious surface and other land-cover types similar to impervious surface (such as cropland and bare land), so the global training samples including several non-impervious land-cover types could be easily collected to build the RF model for accurately mapping of impervious surface. However, as these was temporal interval of 5 years between GlobeLand30 and our study, it was assumed that the process of transforming non-impervious surfaces into impervious surfaces was irreversible during the period 2010 to 2015, meaning that the global impervious training samples derived from GlobeLand30-2010 could also be used to represent the situation in 2015.

Specifically, as GlobeLand30 used an object-based labeling method to remove the "salt-and-pepper effect" caused by the pixel-based classification method (Chen et al., 2015), the impervious surfaces consisted of independent blocks. Usually, a large number of mixed pixels and misclassifications occur at the boundary of image blocks or objects, and Yang et al. (2017) also found that GlobeLand30 exhibited higher accuracy in homogeneous areas. The land-cover heterogeneity was calculated as the number of land-cover types occurring in a local window (Jokar Arsanjani et al., 2016). According to the statistics of Chen et al. (2015), there were a little commission and omission errors in each scene when the area of impervious surface block was less than 8×8. In this study, the local window size was set to 9×9 after balancing the sample reliability and completeness because the higher window size would cause the candidate samples miss those small and broken impervious objects (such as: rural villages). Therefore, if the land-cover heterogeneity in the 9×9 local window was greater than 1 (meaning that the land-cover types within the window consisted of both impervious and non-impervious types), the center pixel was removed from the candidate training point set (CanTPS_Imp).

Secondly, to minimize the effects of mapping error in GlobeLand30-2010 and temporal interval between GlobeLand30-2010 and the input imagery for training samples in CanTPS_Imp, the VIIRS NTL data, revealed the intensity of socioeconomic activities, was imported to refine each training point in 2015. However, as the coarse spatial resolution of VIIRS NTL imagery might cause a 'blooming effect' in suburban areas, the EVI-adjusted night-time light index (EANTLI) proposed by Zhuo et al. (2018) was applied to reduce the blooming effects:

$$EANTLI = \frac{1+(NTL_{norm}-EVI)}{1-(NTL_{norm}-EVI)} \times NTL \, , \tag{1}$$

where $NTL_{norm}$ is the normalized NTL value, $EVI$ is the annual mean value of the time-series MODIS EVI products and $NTL$ is the actual value of the VIIRS NTL data.

The EANTLI measured the likelihood of the pixel corresponding to an impervious surface, so it was reasonable to assume that the pixels where EANTLI exceeded a certain threshold were impervious surface pixels. In this study, as the candidate training points in CanTPS_Imp were collected from homogenous 9×9 pixel areas (270 m×270 m), the EANTLI image in 2015 (EANTLI-2015) was first resampled to the 270 m to match with these candidate points. The GlobeLand30-2010 impervious surface map had a user's accuracy of 86.7%, and we assumed that the process of transforming non-impervious surfaces into

impervious surfaces was irreversible during the period 2010 to 2015, so the impervious segmentation threshold was selected as being the lowest 15th quantile of the cumulative probability of all candidate impervious points for EANTLI-2015; namely, if the cumulative probability of the impervious point in CanTPS_Imp was lower than the threshold, the candidate point was removed from CanTPS_Imp. As for the non-impervious pixels, there was usually a negative correlation between non-impervious surfaces and EANTLI values, and the non-impervious surface samples turned into impervious surface would have high EANTLI values in 2015, so if the cumulative probability of a candidate non-impervious point in CanTPS_Imp was greater than the top 20th percentile of the cumulative probability of all candidate non-impervious points (the threshold being based on the overall accuracy of 80.33% for GlobeLand30-2010 and a little potential conversion samples), the candidate non-impervious point was also removed.

It should be noted that the definition of artificial surfaces in the GlobeLand30 was slightly different from the impervious surfaces in this study. Specifically, artificial surfaces in the GlobeLand30 were divided into three subclasses including high reflectance, low reflectance and vegetated type (Chen et al., 2015), and a small part of purely vegetated artificial surfaces (such as small vegetation patches in the residential zones with good greenness) actually didn't belong to the impervious surfaces. Fortunately, the ENATLI, measuring the likelihood of the pixel corresponding to an impervious surface, usually revealed the low values over these vegetation patches. Therefore, these purely vegetated artificial surface pixels could be removed from the CanTPS_Imp using the lowest 15th quantile of the cumulative probability of all candidate impervious points for EANTLI-2015.

Lastly, although the candidate training points were refined using the GlobeLand30 land-cover product and EANTLI-2015 imagery, the volume of candidate training points was still huge and so it was necessary to further resample the CanTPS_Imp. As the non-impervious surfaces consisted many land-cover types (water, vegetation, cropland and bare soil) and some of them were spectrally similar to the impervious surface. For example, the bare soil and high reflectance impervious surfaces usually shared similar surface reflectance especially in arid and semi-arid areas with large areas of bare soils because the composition of impervious surfaces included rock material which was also found in bare areas (Sun et al., 2019b; Weng, 2012), the cropland showed similar reflectance to these low reflectance impervious surfaces (such as rural village, old cities) because they were usually composited of vegetation and high reflectance artificial materials or bare soils (Li et al., 2015). Therefore, the non-impervious training samples were split into three independent groups including: bare area, cropland and other non-impervious land-cover types. Furthermore, many studies had demonstrated that the distribution and balance of training samples had great influence on the mapping accuracy. For example, Zhu et al. (2016) found unbalanced training samples directly resulted in rare land-cover types under-represented relative to more abundant classes. Since the impervious surface was usually sparser than the non-impervious land-cover types (bare soil, cropland and so on), the training samples with uniform distribution were selected to ensure the rationality of training samples and capture all relevant spectral heterogeneity within impervious surfaces, namely, the approximate ratio of 1:3 was used to represent the proportion of impervious to non-impervious surfaces (bare area, cropland and other non-impervious land-cover types). In addition, as the land-cover distribution varied with geographical

region, the stratified random sampling strategy was applied at every 5 °×5 ° geographical grids to ensure the training samples locally adaptive. Using the stratified random sampling strategy with the uniform distribution, a total of 4,483,000 training samples, including 3,499,000 non-impervious samples and 984,000 impervious samples, were collected over the land areas across the globe.

Although a series of rules were applied to guarantee the high confidence of global training samples, due to the classification error in GlobeLand30 and the temporal interval between GlobeLand30 and input imagery, the global training dataset inevitably contained some erroneous samples. The relationship between the percentage of the erroneous samples and the mapping accuracy of impervious surface was analyzed in the Discussion section 6.1, and the results indicated that the error in the training samples had little effect on the mapping accuracy.

## 4 Multi-source and multi-temporal impervious classification method

To develop the global 30-m impervious surface map for 2015, the MSMT_RF method was proposed. The method is illustrated in Fig. 2. First, time series of Landsat-8 SR and Sentinel-1 SAR imagery archived on the GEE platform were collected. Secondly, the temporal−spectral−textural features and temporal−SAR features were derived from the Landsat-8 and Sentinel-1 imagery using image compositing methods. Thirdly, based on the global training samples derived from GlobeLand30-2010, VIIRS NTL and MODIS EVI imagery, the random forest classifier was trained at each 5 °×5 ° geographical grid cell using the temporal−spectral−textural−SAR-topographical features. Finally, the global impervious surface map was compared with existing impervious surface products and further validated using the visual interpretation samples.

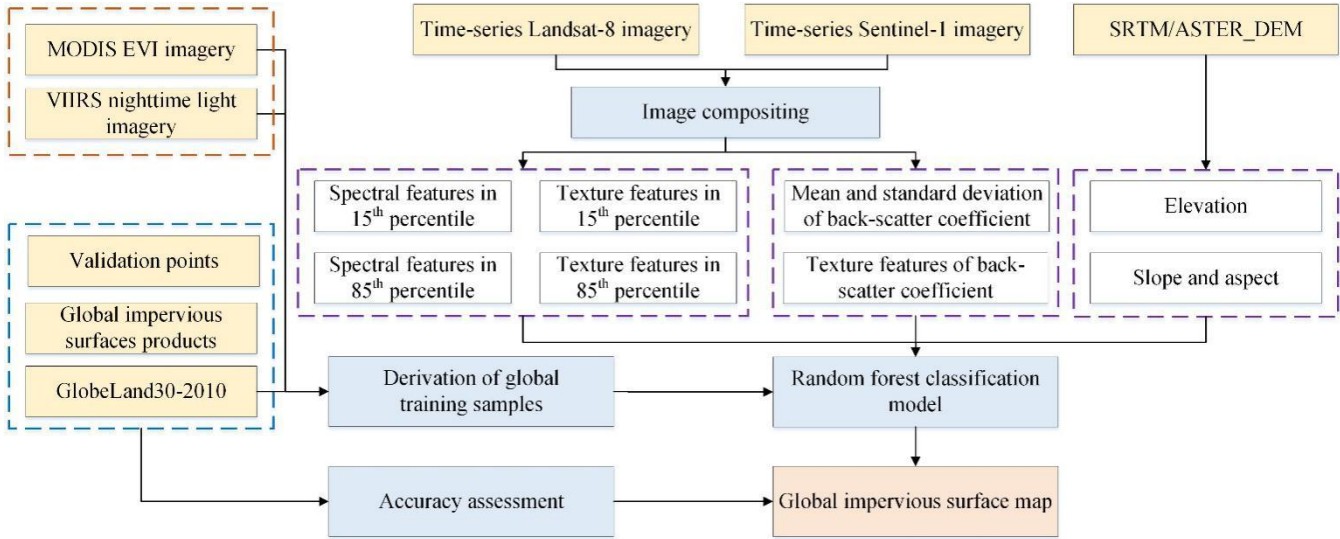

**Figure 2: Flowchart illustrating the MSMT_RF method.**

## 4.1 Multi-source and multi-temporal feature selection

As mentioned above, the datasets used in this study had been acquired from various satellite sensors and had distinctive features. Also the incorporation of multi-source and multi-temporal remote sensing data has been demonstrated to improve the accuracy of the mapping of impervious surfaces. In this study, three kinds of satellite imagery, including Landsat-8 SR, Sentinel-1 SAR and SRTM/ASTER DEM imagery, were collected for the global classification of impervious surfaces.

After masking out the bad pixels (cloud, shadow and saturated pixels), the time-series Landsat SR imagery were needed to reduce the number of dimensions of the temporal−spectral features to guard against the Hughes phenomenon (Zhang et al., 2019). Similar to what Hansen et al. (2014) and Zhang and Roy (2017) introduced to capture phenology, the 15th and 85th percentiles of Landsat SR were used instead of the minimum and maximum values to minimize the effects of residual shadows and cloud caused by the errors in the CFMask method (Massey et al., 2018). In addition, as Sun et al. (2017) explained that the growing season was the best time for impervious surface mapping over temperate continental climate zones and Zhang et al. (2014a) found that winter (dry season) is the best season to estimate impervious surface in subtropical monsoon regions, the combination of 15th and 85th percentiles of Landsat SR was used to efficiently capture intra-annual variation information of various land-cover types. It should be noted that only the six optical bands (Blue, Green, Red, NIR, SWIR1 and SWIR2) were selected because the Coastal band was sensitive to the atmospheric scattering (Wang et al., 2016). Liu et al. (2018) found that the Normalized Difference Water Index (NDWI), Normalized Difference Vegetation Index (NDVI) and Normalized Difference Built-up Index (NDBI) were of great help in impervious surface identification; therefore, these three spectral indexes were added to the spectral features, giving a total of 18 features for the two-epoch imagery. Furthermore, as the texture information contributed to the classification performance (Weng, 2012), the local textural measures based on the Gray Level Co-occurrence Matrix (GLCM) were adopted; however, because of the redundancy and similarity between texture features (Rodriguez-Galiano et al., 2012), only the variance, dissimilarity and entropy of the NIR band were selected from the 7×7 local window for the two-epoch imagery (Chen et al., 2016; Zhang et al., 2014b). The optimal window size for texture measurements was highly dependent on the image spatial resolution and the land cover characteristics (Zhu et al., 2012) and Shaban and Dikshit (2001) computed texture measurements with different window sizes as inputs for urban area classification and suggested window sizes of 7 ×7 pixels perform best.

As the Sentinel-1 SAR imagery had been pre-processed in the GEE platform, the annual mean and standard deviation of the VV and VH imagery were directly derived from the time-series of Sentinel-1 SAR imagery. Zhang et al. (2014b) found that SAR texture features were also relevant to impervious surfaces and the dissimilarity, variance and entropy features of the VV and VH imagery were identified as effective indicators for the texture description of different urban land cover types. As Zhang et al. (2014b) explained the window size for calculating GLCM should be smaller as terrains are smaller under coarser resolution, the window size was chose as 9×9 pixels at 10-m spatial resolution, equivalent to 3×3 pixels in 30-m. Moreover, as the spatial resolution of the Landsat SR (30-m) was three times that of the Sentinel-1 imagery (10-m), the SAR data were resampled to 30-m for integration with the Landsat SR data.

Lastly, as Sentinel-1 SAR imagery usually had high backscatter similar to the impervious surface over mountainous areas, terrain information were useful auxiliary for removing these false positive at these areas (Ban et al., 2015). Similarly, Clarke et al. (1997) found that terrain variables were of great help in identifying impervious surfaces because they usually located in the flat areas. In this study, the elevation, slope and aspect, calculated from the SRTM/ASTER DEM data, were added to the feature vector. This gave a total of 37 features for each pixel location, including 18 spectral features and 6 texture features from the Landsat imagery, 10 SAR features and 3 topographical variables. The features are listed in Table 1.

**Table 1. Training features for global impervious surface mapping.**

| Data | Features | References |
|---|---|---|
| LandSat-8 OLI | Reflectance: 15th and 85th percentiles of Blue, Green, Red, NIR, SWIR1 and SWIR2 | Liu et al. (2018) |
| | Normalized indices: 15th and 85th percentiles of NDVI, NDWI and NDBI | |
| | Textural variables: variance, dissimilarity and entropy of the NIR | Chen et al. (2016) |
| Sentinel-1 SAR | Annual statistics: mean and standard deviation of VV and VH | Sun et al. (2019b) |
| | Textural features: dissimilarity, variance and entropy of VV and VH | Zhang et al. (2014b) |
| DEM | Elevation, slope and aspect | Clarke et al. (1997) |

## 4.2 Random forest classification model

There were two kinds of models used for generating a global impervious surface product – global modeling (building a single classifier using global training data) and local adaptive modeling (dividing the globe into a number of regions and then building local classifiers using corresponding regional training data). For example, Gong et al. (2013) built a single global classifier using 91,433 training samples to produce the FROM_GLC land-cover products; Bontemps et al. (2011) first split the world into 22 ecological regions and then trained the classifier for each region using local training samples to produce the GlobeCover2009 land-cover products. Recently, Zhang and Roy (2017) demonstrated that the local adaptive model performed better than the single global classification model. While Radoux et al. (2014) found that using a local window increased the sensitivity to the quality of the training dataset. Therefore, after balancing the data volume, computation efficiency and classification accuracy, we first split the global land surface into 954 5 °×5 °geographical tiles and then trained local adaptive classifiers for each geographical tile. In addition, to ensure the classification consistency across neighboring geographical tiles, the training data from adjacent $3 \times 3$ tiles were imported to train the random forest classifier and classify the central tile.

As for the specific techniques used in classifiers, according to our previous investigations (Zhang et al., 2019), the Random Forest classifier is more capable of handling high-dimensional multicollinearity data. It is also less affected by noise and feature selection as well as being more accurate and efficient than other widely used classifiers such as the SVM (Support Vector Machine), CART (Classification And Regression Tree) and ANN (Artificial Neural Network) classifiers. Therefore, the RF classifier was selected for the development of the global impervious surface map.

The RF classifier has only two parameters: the number of classification trees (Ntree) and the number of selected predication features (Mtry). Furthermore, many researchers have demonstrated that there is almost no correlation between these two parameters and the classification accuracy (Belgiu and Drăguţ, 2016; Du et al., 2015; Gislason et al., 2006); therefore, the default values of 500 for Ntree and the square root of the total number of training features for Mtry were selected.

### 4.3 Accuracy assessment

To completely analyze the performance of the MSMT_RF-based method, two validation methods including 'fraction-based' and 'pixel-based' were adopted. First, the 'fraction-based' validation method mainly illustrated the spatial agreement of impervious surfaces between the MSMT_RF-based impervious surface map and several existing products (GlobeLand30-2010, FROM_GLC-2015, NUACI-2015, HBASE-2010 and GHSL-2015) from a global perspective. Specifically, all these global 30-m impervious surface maps were aggregated to a resolution of $0.05°\times0.05°$ and the fraction of impervious area was then calculated. Following that, scatter plots of the linear regression between the MSMT_RF-based results and the reference data were produced to provide the quantitative metrics of the agreement, including coefficient of determination ($R^2$) and root mean square error (RMSE).

In addition, a 'pixel-based' validation method, based on the visual interpretation samples over fifteen $1°\times1°$ regions covering different impervious landscapes and continents, was used to quantitatively analyze the accuracy metrics, including overall accuracy (O.A.), producer's accuracy (P.A.), user's accuracy (U.A.) and kappa coefficient (Olofsson et al., 2014) for assessing the performance of the MSMT_RF-based global impervious surface mapping.

## 5 results

### 5.1 The importance of multi-source and multi-temporal features

Because of the spectral heterogeneity of impervious surfaces, it is very difficult to accurately map impervious surfaces using only optical remote sensing imagery (Zhang et al., 2014b). Although a few studies have demonstrated that the integration of multi-source and multi-temporal information can improve the mapping accuracy, these studies mainly focused on regions with high impervious surface density (Zhang et al., 2014b; Zhu et al., 2012). At present, global impervious surface maps are still produced by optical imagery alone or by using a combination of optical and DMSP-OLS or VIIRS NTL imagery (Huang et al., 2016; Liu et al., 2018; Schneider et al., 2010). This is the first study that developed the global 30-m impervious surface map using multi-source and multi-temporal imagery. To quantitatively demonstrate the need for using multi-source, multi-temporal information, we randomly selected six $5°\times5°$ regions (red rectangles in Fig. 1) from six different continents and then calculated the importance of the training features using the RF model. Specifically, the RF model computed the average increase in the mean square error by permuting out-of-bag data for a variable while keeping all the other variables constant, thus measuring the variable's importance (Pflugmacher et al., 2014). Training features that had a high importance were the drivers of the model decision and their values had a significant impact on the output values.

The importance of all 37 training features for the six regions is illustrated in Fig.3. These results indicate that the Sentinel-1 SAR features (VV and VH) had the greatest contribution to the final decision in most regions because SAR images can provide information about the structure and dielectric properties of the surface materials. Next in importance were the 15th percentile of Landsat SR in the blue, green, red and SWIR2 bands and the corresponding NDVI and NDWI indices, as well as the texture variance and dissimilarity for Sentinel-1 SAR. The importance of these feature was close to or exceeded 5% in most cases. Then came the 85th percentile of Landsat SR in the NIR and SWIR1 bands as well as the SAR texture features, with a mean importance about 3%.

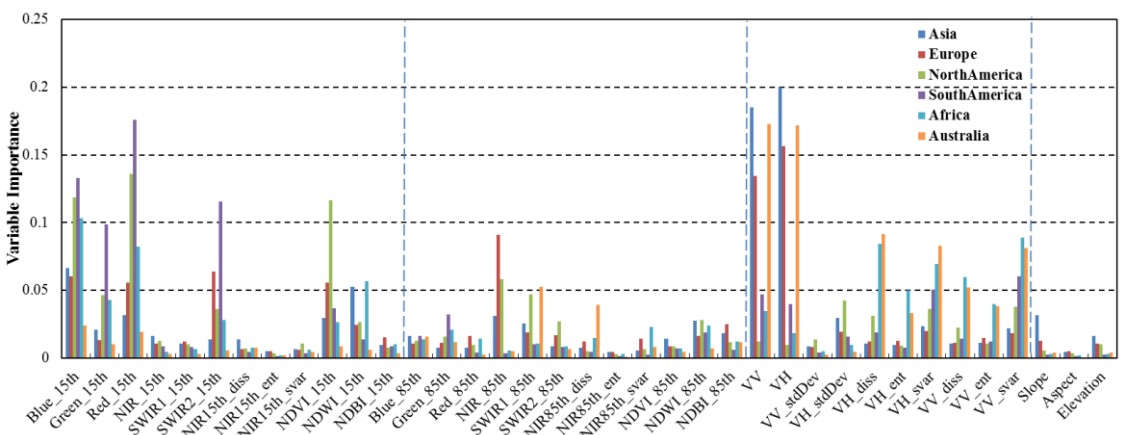

**Figure 3: The importance of the input features derived from the random forest model using the training samples in six continental regions.**

To intuitively understand the characteristics of different land-cover types on optical and SAR imagery, two regions (the vegetation-prevalent region of Asia and bare soil-prevalent semi-arid region of Australia) were selected for comparison analysis. Fig. 4 illustrated the reflectance and backscatter statistics (mean and standard deviation) of five typical land-cover types (cropland, vegetation, bare soil, impervious surfaces and water body). Obviously, impervious surfaces had highest backscatter signals in VV because of the high dielectric properties of the building materials, the unique geometry of manmade features, and the special radar echo properties of artificial structures, followed by the vegetation land-cover types. Further, since only a small part of the polarized signals (vertical turning horizontal) were returned to the sensor, the VH was significantly lower than VV but the ranking orders of different land-cover types in VH was similar to that of VV. Due to the complicated construction and heterogeneity of the impervious surfaces, the impervious surfaces also had highest standard deviation, for example, the urban central usually reflected higher VV and VH signals than the village buildings. If only Sentinel-1 SAR features were used to identify impervious surfaces, there would be serious confusion between the mountainous vegetation with low reflectance impervious surfaces (such as: villages and small cities), fortunately, the optical reflectance features performed well to distinguish them because of significant spectral differences. However, if only the multi-temporal optical imagery were used to detect the impervious surfaces, there would be obvious confusion between impervious surfaces with bare soils and croplands, for example, the spectral characteristics of impervious surfaces, bare soils and croplands were

overlapping in the Asia region (Fig. 4). In summary, only the combination of multi-source training features could guarantee the classification accuracy across different impervious landscapes.

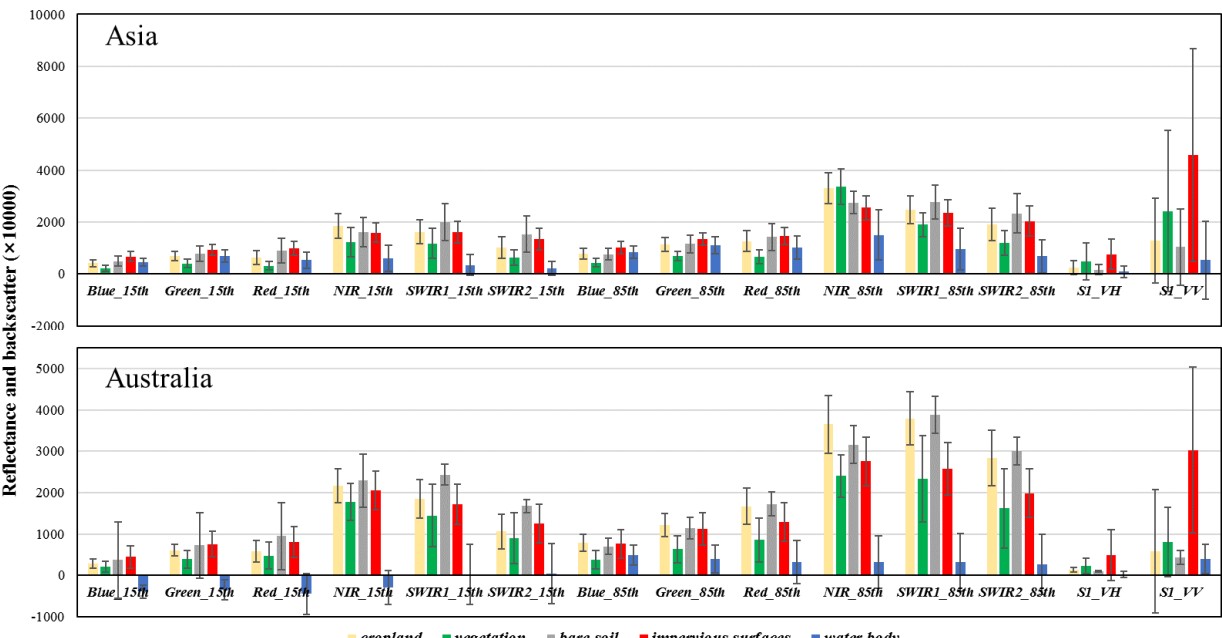

**Figure 4: The reflectance/backscatter characteristics of different land-cover types over Landsat optical and Sentinel-1 SAR imagery in the Asia and Australia regions.**

Secondly, although the 15th percentile had a higher importance than the 85th percentile in most of the spectral bands, we found that there was a large degree of complementarity between the images from two different seasons (Fig. 3). For example, the importance of the 15th percentile in the NIR and SWIR1 bands was low while that of 85th percentile was high, and the total importance of the bi-seasonal spectral features exceeded 70% in some cases. The reasons that the temporal information was important for accurately mapping of impervious surface included: (1) some land-cover types such as cropland had similar spectra with impervious surface at fallow season, but with the growing season imagery imported, this misclassification could be easily removed; (2) Sun et al. (2017) explained that the growing season was the best time for impervious surface mapping over temperate continental climate zones, and Zhang et al. (2014a) found that winter (dry season) is the best season to estimate impervious surface in subtropical monsoon regions. The multi-temporal information can address the problem of seasonal variability at different geographical zones. Fig. 4 (Australia region) also illustrated that the cropland and impervious surfaces were spectrally inseparable in the 15th percentile but the difference was obvious in the 85th percentile. Therefore, temporal variability can be considered an important contribution for accurate impervious surface mapping.

Thirdly, the importance of Landsat texture features was lower than 5% in these six regions because the Sentinel-1 SAR backscatter and texture features were able to provide information on the surface material and its spatial structure and variation. Due to the complexity of land-surfaces and different mechanism of optical and SAR imagery, the optical textures could

complement a lot to SAR features at mountainous and semiarid areas (Asia and Australia regions). Some studies demonstrated
that these features contributed a lot to the improvement of impervious mapping accuracy. For example, Shaban and Dikshit (2001) emphasized that the integration of texture variables increased the accuracy from 86.86% to 92.69% because texture imagery could capture the local spatial structure and the variability of land-cover categories.

Lastly, since most regions are located in the flat areas, only the cumulative importance of topographical variables over the region in Asia exceeded 5%. The reasons why topographical information reached high importance over mountainous areas
were because the impervious surfaces usually located in the flat areas (Ban et al., 2015) and Sentinel-1 SAR imagery had high backscatter signals over mountainous areas similar to the impervious surfaces, which increased the importance of topographical variables. Similarly, Clarke et al. (1997) explained that topographical variables (slope, aspect and DEM) contribute a lot to impervious surface mapping over mountainous areas. These features are, therefore, indispensable in the accurate mapping of impervious surfaces in mountainous regions.

**5.2 Global impervious surface map**

The global distribution of the fraction of impervious area (FIA) at a spatial resolution of 0.05 ° is illustrated in Fig. 5, whilst the meridional and zonal total FIA for each 0.05 ° longitude and latitude bin are shown at the top and left of Fig. 3. From an intuitive and statistical perspective, globally, impervious surfaces are mainly concentrated in three continents: Asia (34.43%), North America (28.04%) and Europe (24.98%), followed by South America (5.89%), Africa (5.63%) and Australia (1.06%).
In addition, the zonal statistics indicate that 70% of the impervious surfaces are distributed between 30 ˚N and 60 ˚N because these regions contain the key areas of Asia, North America and Europe, which are the locations of the most developed countries and highest population densities. The meridional results illustrate that there are four peak intervals: 100 °W to 70 °W (United States), 10 °W to 40 °E (European Union), 60 °E to 90 °E (India) and 100 °E to 130 °E (China and southeastern Asia). The two peak values in the meridional direction are located in the centers of the United States and China.

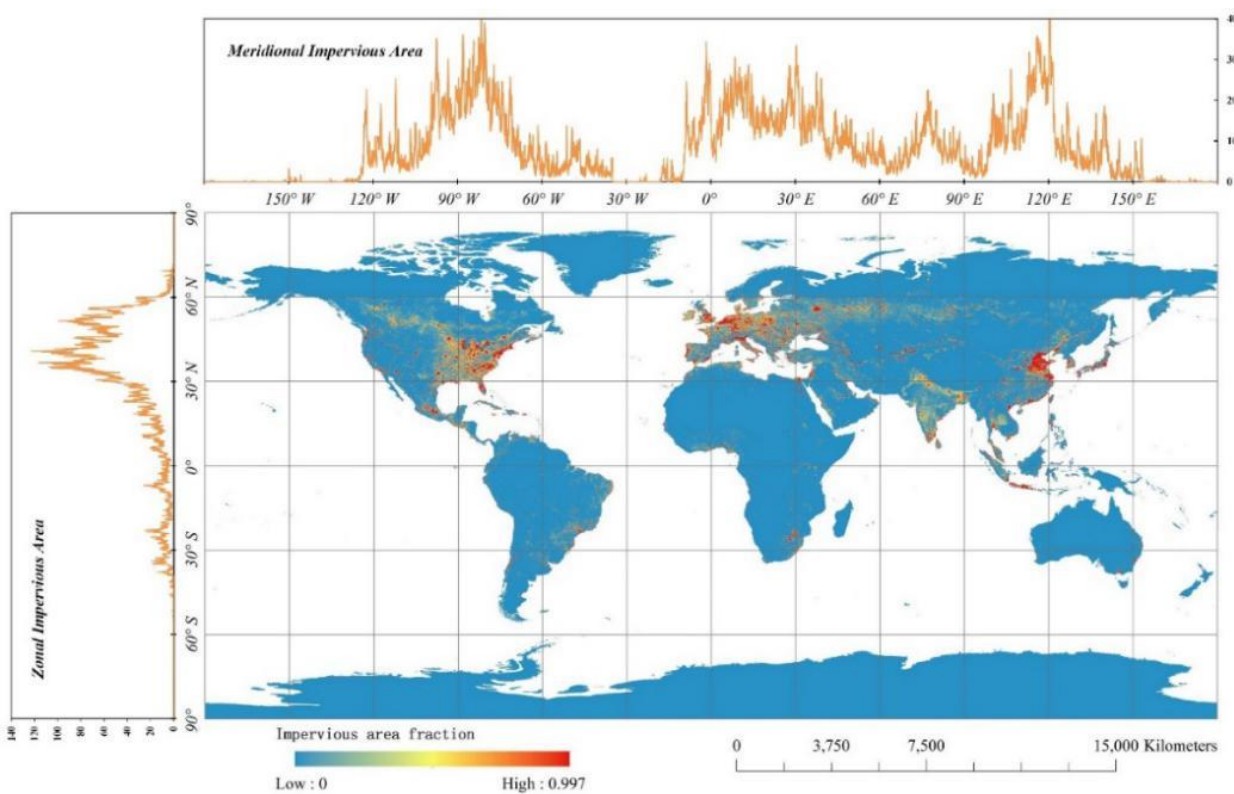


**Figure 5: Global fraction of impervious cover derived from multi-source and multi-temporal Landsat-8 SR and Sentinel-1 SAR imagery acquired in 2015 to 2016. The spatial resolution of the map is 0.05 °.**

Summaries of the impervious surface areas at a national scale were also produced. The statistical results indicated that the total impervious surface areas of the top 20 countries account for 75.96% of the total global area. Fig. 6 presents the top 20 counties

in terms of impervious surface area and corresponding fractions of the world total. Overall, there is a positive correlation between these statistical fractions and the land area, population and degree of economic development of these nations. Specifically, it was found that the U.S. has the biggest impervious surface area, accounting for more than 20% of the global total, and only the top 3 countries (U.S., China and Russia) exceed 5% of the total global area. The ranking is also basically consistent with the statistics produced by the Organization for Economic Co-operation and Development (OECD) for built-up

areas in 2014 (https://stats.oecd.org/Index.aspx?DataSetCode=BUILT_UP).

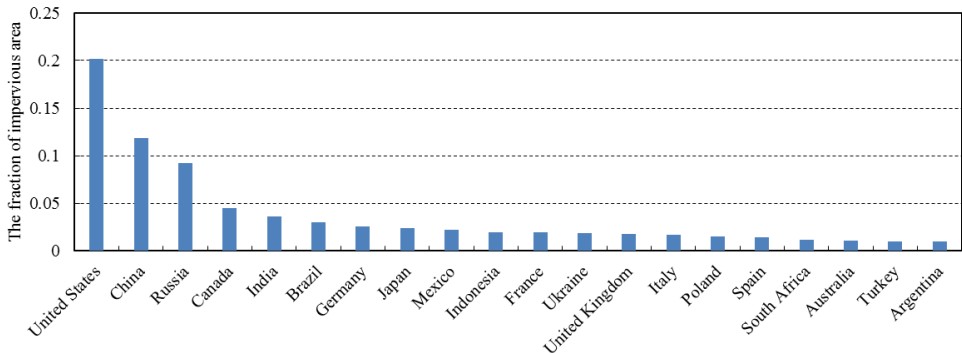

**Figure 6: The top 20 countries in terms of impervious surface area and corresponding fractions of the global total.**

## 5.3 Spatial variations of global impervious products

To quantitatively analyse the spatial agreement between the MSMT_RF-based impervious surface map and the five existing
products (GlobeLand30-2010, FROM_FLC-2015, NUACI-2015, GHSL-2015 and HBASE-2010), all global 30-m impervious
surface maps were first aggregated to a resolution of 0.05 °. Fig. 7 illustrated the spatial patterns of six global impervious
products, intuitively, the NUACI-2015 had lower impervious areas than other products especially in the North-America and
Europe, and the GHSL-2015, GlobeLand30-2010 and our product (MSMT-2015 map) had greater spatial agreement because
the impervious areas of FROM_GLC-2015 and HBASE-2010 in the China were obviously smaller. Further, our product had
higher impervious areas over North-America especially over the Canada than other products because the proposed method had
greater ability to identify small and fragmented impervious objects such as villages and roads which was been demonstrated
in the following section 5.4 over Winnipeg region.

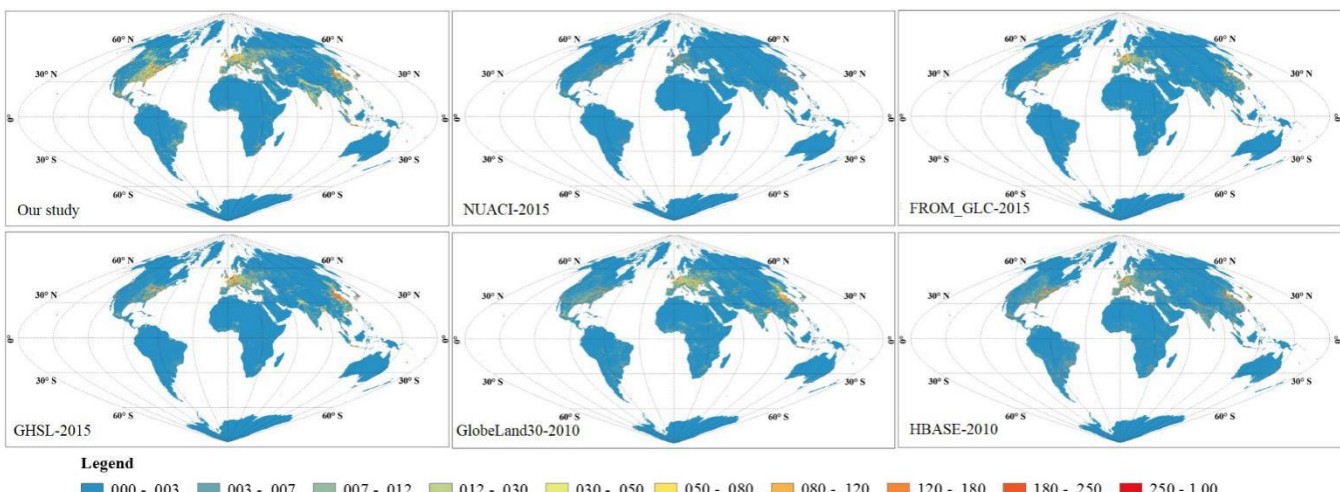

**Figure 7: The spatial patterns of six global 30-m impervious products after aggregating to the resolution of 0.05 °.**

Scatter plots of the five products against the MSMT-2015 impervious map were then made, as illustrated in Fig. 8. The results
indicate that there was a greater agreement between the MSMT-2015 map and GHSL-2015 ($R^2$=0.783, RMSE= 0.038 and

slope=0.921) than for other products. Specifically, as NUACI-2015 has been demonstrated to miss some small, fragmented villages and roads (Sun et al., 2019b), the slope of the regression line was less than 1.0 and R2 was the low value of 0.655 in this case. The scatter plot between FROM_GLC-2015 and MSMT-2015 indicated that there was a high degree of agreement

between FROM_GLC-2015 and MSMT-2015 results in 'high-fraction' regions (close to 1:1) but FROM_GLC-2015 was obviously lower than MSMT-2015 over 'low fraction' regions, so the slope of the regression line for FROM_GLC-2015 was also less than 1. The main differences between the GlobeLand30 and the MSMT_RF-based maps were due to the temporal interval of 5 years and the limitations of the minimum 4×4 mapping unit for GlobeLand30-2010 (Chen et al., 2015), so the scatters were mainly concentrated below the 1:1 line. The HBASE-2010 had higher impervious areas than MSMT-2015

especially for the 'high-fraction' regions, but the following section demonstrated that it suffered the over-estimation problem, so the regression slope was higher than 1 and $R^2$ only reached the value of 0.730. In addition, to intuitively understand the stability of regression model, the error bars, calculated as the standard deviation of reference data with the fitted results, were added to the scatter plots. It could be found that the error bars increased first and then stabilized as the impervious fraction increased.

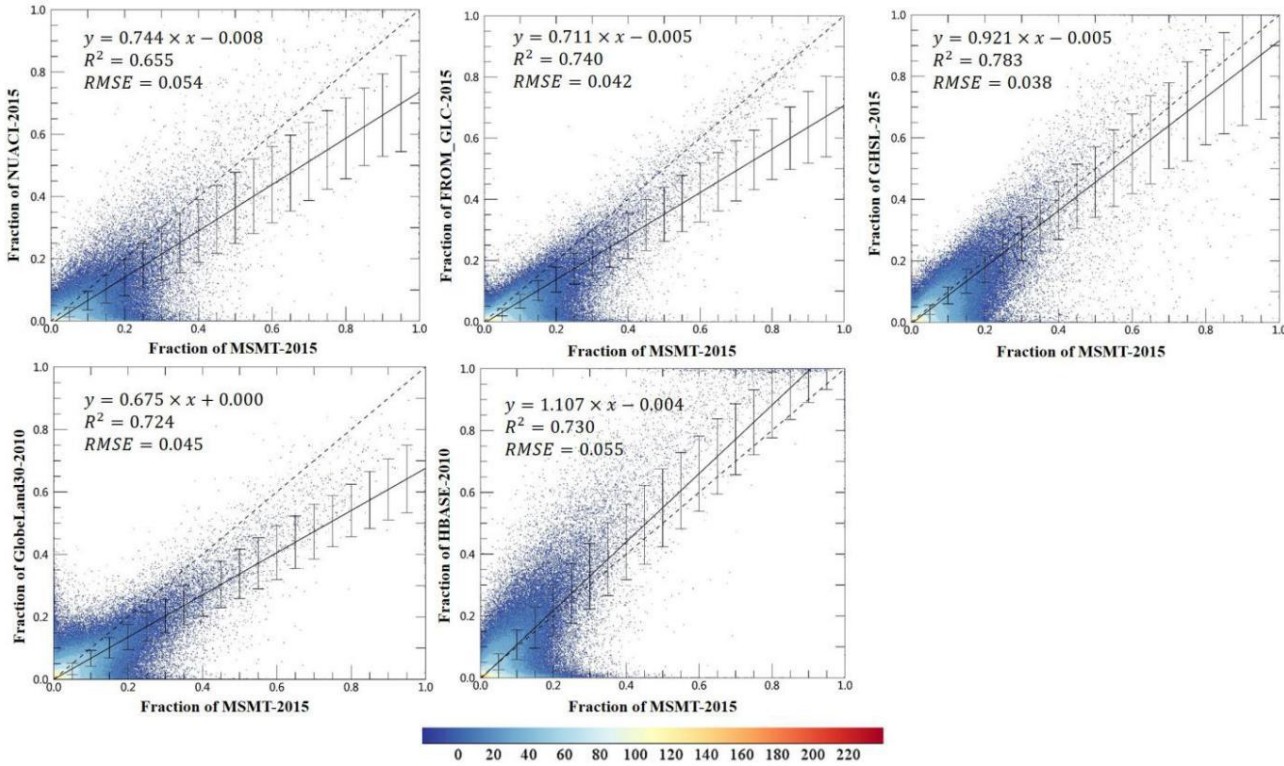


**Figure 8: Scatter plots between the MSMT_RF-based impervious map and the GlobeLand30-2010, FROM_GLC-2015, NUACI-2015, GHSL-2015 and HBASE-2010 global impervious surface products at a spatial grid of 0.05°×0.05°. The error bars were the standard deviation between reference datasets with fitted results.**

## 5.4 Accuracy assessment using validation samples

The accuracy of the five global impervious surface maps over 15 validation regions with different impervious landscapes is presented in Table 2. Six evaluation metrics, including the producer's accuracy (which measures the commission error) and user's accuracy (which measures the omission error) of the impervious surface, the producer's and user's accuracy of non-impervious surfaces as well as the overall accuracy and kappa coefficient, were used to assess the accuracy. Overall, the MSMT_RF-based map achieved the highest overall accuracy of 0.951 and kappa coefficient of 0.898 compared with 0.896 and 0.780 for FROM_GLC-2015, 0.856 and 0.695 for NUACI-2015, 0.903 and 0.794 for GHSL-2015, 0.884 and 0.753 for GlobeLand30-2010, and 0.880 and 0.754 for HBASE-2010 using all 15 regional validation data.

From the perspective of the value of the user's accuracy for impervious surfaces, the MSMT_RF method performed better than the other impervious surface products (meaning lower omission error) achieving the accuracy of 0.932, especially in the cropland-prevalent and vegetation-prevalent impervious landscapes (such as: Bangkok, Winnipeg, Xi'an…). Specifically, NUACI-2015 had the lowest user's accuracy of 0.562 and this might be due to its poor performance over small impervious surfaces (Sun et al., 2019b). FROM_GLC-2015 had a similar performance with the MSMT_RF method for big cities (such as New York, Moscow and Johannesburg), but its accuracy decreased sharply over 'small-city' regions (such as Lhasa, Winnipeg). The performance of GHSL-2015 was closest to the MSMT-2015 over most validation regions, but it also missed the fragmented objects (villages and roads) over cropland-prevalent city (such as Bangkok and Winnipeg). As the minimum mapping unit of GlobeLand30 was a 4×4-pixel area, many rural impervious surfaces were ignored in these validation regions, which caused large omission errors of 23.9%. Finally, partly due to the 5 years' interval between the HBASE-2010 and validation samples, HBASE-2010 also suffered the omission error of 12.5%.

As for the producer's accuracy for impervious surface (measuring the commission error), the GHSL-2015 products performed best and achieved the accuracy of 0.973, followed by the MSMT-2015 of 0.948, GlobeLand30-2010 of 0.947, FROM_GLC-2015 of 0.946, NUACI-2015 of 0.898 and HBASE-2010 of 0.841. Compared with user's accuracy of impervious surface, these reference products had better performance on this metric, which meant they had lower commission error.

**Table 2. Accuracy of the six impervious surface maps over 15 validation regions**

| | NAME | BGK | JHB | LHS | MDR | MNS | MBN | MSC | NYK | NIM | NTU | PNX | RYH | SPL | WIP | XAN | O.A. |
|---|---|---|---|---|---|---|---|---|---|---|---|---|---|---|---|---|---|
| | **I.L.** | CR | BS | BS | BS | VG | VG | VG | VG | BS | BS | BS | BS | VG | CR | CR | |
| MSMT-2015 | **U.I.** | 0.951 | 0.963 | 0.691 | 0.929 | 0.993 | 0.957 | 0.987 | 0.995 | 0.869 | 0.750 | 0.988 | 0.918 | 0.984 | 1.000 | 0.929 | 0.932 |
| | P.I. | 0.997 | 0.922 | 0.989 | 0.961 | 0.938 | 0.972 | 0.961 | 0.981 | 0.987 | 0.951 | 0.975 | 0.944 | 0.965 | 0.915 | 0.940 | 0.948 |
| | U.N. | 0.997 | 0.958 | 0.996 | 0.986 | 0.966 | 0.987 | 0.949 | 0.952 | 0.997 | 0.975 | 0.975 | 0.954 | 0.978 | 0.958 | 0.922 | 0.964 |
| | P.N. | 0.951 | 0.981 | 0.873 | 0.975 | 0.996 | 0.980 | 0.982 | 0.987 | 0.964 | 0.859 | 0.987 | 0.932 | 0.990 | 1.000 | 0.909 | 0.953 |
| | O.A. | 0.974 | 0.960 | 0.899 | 0.971 | 0.975 | 0.978 | 0.970 | 0.983 | 0.969 | 0.888 | 0.981 | 0.938 | 0.980 | 0.971 | 0.926 | 0.951 |
| | **Kappa** | 0.948 | 0.912 | 0.747 | 0.925 | 0.945 | 0.948 | 0.939 | 0.957 | 0.904 | 0.754 | 0.963 | 0.874 | 0.958 | 0.934 | 0.850 | 0.898 |
| NUACI | **U.I.** | 0.695 | 0.885 | 0.031 | 0.469 | 0.935 | 0.690 | 0.933 | 0.960 | 0.526 | 0.587 | 0.765 | 0.822 | 0.935 | 0.777 | 0.562 | 0.735 |

| | | | | | | | | | | | | | | | | | |
|---|---|---|---|---|---|---|---|---|---|---|---|---|---|---|---|---|---|
| | P.I. | 0.979 | 0.693 | 0.889 | 0.818 | 0.952 | 0.918 | 0.977 | 0.927 | 0.968 | 0.915 | 0.968 | 0.912 | 0.917 | 0.923 | 0.927 | 0.898 |
| | U.N. | 0.985 | 0.800 | 0.998 | 0.963 | 0.975 | 0.970 | 0.972 | 0.788 | 0.995 | 0.965 | 0.975 | 0.933 | 0.947 | 0.971 | 0.943 | 0.941 |
| | P.N. | 0.757 | 0.932 | 0.686 | 0.835 | 0.966 | 0.868 | 0.919 | 0.884 | 0.882 | 0.785 | 0.806 | 0.862 | 0.959 | 0.907 | 0.624 | 0.834 |
| | O.A. | 0.838 | 0.829 | 0.689 | 0.833 | 0.961 | 0.880 | 0.950 | 0.911 | 0.893 | 0.818 | 0.870 | 0.883 | 0.943 | 0.911 | 0.728 | 0.856 |
| | **Kappa** | 0.677 | 0.641 | 0.040 | 0.500 | 0.914 | 0.706 | 0.899 | 0.789 | 0.624 | 0.590 | 0.740 | 0.761 | 0.879 | 0.783 | 0.476 | 0.695 |
| FROM_GLC-2015 | **U.I.** | 0.717 | 0.952 | 0.027 | 0.844 | 0.938 | 0.891 | 0.953 | 0.984 | 0.549 | 0.763 | 0.883 | 0.749 | 0.935 | 0.854 | 0.595 | 0.794 |
| | P.I. | 0.990 | 0.779 | 1.000 | 0.973 | 0.974 | 0.958 | 0.982 | 0.972 | 0.960 | 0.930 | 1.000 | 0.975 | 0.986 | 0.981 | 0.982 | 0.946 |
| | U.N. | 0.992 | 0.862 | 1.000 | 0.992 | 0.987 | 0.982 | 0.977 | 0.931 | 0.994 | 0.963 | 1.000 | 0.984 | 0.992 | 0.993 | 0.986 | 0.968 |
| | P.N. | 0.772 | 0.972 | 0.686 | 0.947 | 0.968 | 0.950 | 0.942 | 0.960 | 0.887 | 0.864 | 0.895 | 0.823 | 0.961 | 0.938 | 0.652 | 0.870 |
| | O.A. | 0.853 | 0.893 | 0.689 | 0.953 | 0.970 | 0.953 | 0.964 | 0.969 | 0.896 | 0.885 | 0.941 | 0.876 | 0.970 | 0.950 | 0.765 | 0.896 |
| | **Kappa** | 0.706 | 0.772 | 0.037 | 0.872 | 0.933 | 0.889 | 0.927 | 0.923 | 0.641 | 0.750 | 0.883 | 0.746 | 0.936 | 0.879 | 0.548 | 0.780 |
| GHSL-2015 | **U.I.** | 0.619 | 0.752 | 0.453 | 0.815 | 0.880 | 0.849 | 0.958 | 0.991 | 0.451 | 0.619 | 0.940 | 0.672 | 0.925 | 0.899 | 0.741 | 0.787 |
| | P.I. | 1.000 | 0.949 | 1.000 | 0.989 | 0.996 | 0.978 | 0.982 | 1.000 | 1.000 | 1.000 | 0.995 | 0.996 | 0.996 | 0.991 | 0.968 | 0.973 |
| | U.N. | 1.000 | 0.979 | 1.000 | 0.997 | 0.998 | 0.991 | 0.977 | 1.000 | 1.000 | 1.000 | 0.995 | 0.998 | 0.998 | 0.996 | 0.968 | 0.985 |
| | P.N. | 0.717 | 0.886 | 0.795 | 0.938 | 0.941 | 0.932 | 0.948 | 0.979 | 0.867 | 0.804 | 0.943 | 0.783 | 0.955 | 0.957 | 0.742 | 0.868 |
| | O.A. | 0.806 | 0.903 | 0.825 | 0.949 | 0.958 | 0.945 | 0.966 | 0.994 | 0.880 | 0.851 | 0.968 | 0.849 | 0.970 | 0.966 | 0.840 | 0.903 |
| | **Kappa** | 0.615 | 0.770 | 0.530 | 0.860 | 0.903 | 0.870 | 0.932 | 0.985 | 0.563 | 0.664 | 0.935 | 0.687 | 0.936 | 0.919 | 0.686 | 0.794 |
| GlobeLand30-2010 | **U.I.** | 0.310 | 0.704 | 0.410 | 0.825 | 0.804 | 0.744 | 0.908 | 0.981 | 0.537 | 0.779 | 0.923 | 0.831 | 0.902 | 0.749 | 0.750 | 0.761 |
| | P.I. | 0.992 | 0.950 | 0.991 | 0.978 | 0.961 | 0.975 | 0.962 | 0.954 | 1.000 | 0.968 | 0.966 | 0.905 | 0.972 | 0.954 | 0.874 | 0.947 |
| | U.N. | 0.997 | 0.981 | 0.998 | 0.993 | 0.983 | 0.991 | 0.955 | 0.901 | 1.000 | 0.984 | 0.968 | 0.926 | 0.984 | 0.984 | 0.859 | 0.970 |
| | P.N. | 0.582 | 0.867 | 0.782 | 0.941 | 0.905 | 0.891 | 0.891 | 0.955 | 0.885 | 0.874 | 0.926 | 0.866 | 0.942 | 0.898 | 0.726 | 0.852 |
| | O.A. | 0.648 | 0.888 | 0.810 | 0.949 | 0.921 | 0.911 | 0.929 | 0.936 | 0.899 | 0.904 | 0.945 | 0.883 | 0.953 | 0.911 | 0.798 | 0.884 |
| | **Kappa** | 0.303 | 0.731 | 0.483 | 0.861 | 0.818 | 0.783 | 0.857 | 0.917 | 0.645 | 0.790 | 0.890 | 0.762 | 0.898 | 0.779 | 0.597 | 0.753 |
| HBASE-2010 | **U.I.** | 0.801 | 0.915 | 0.527 | 0.888 | 0.913 | 0.744 | 0.984 | 0.998 | 0.720 | 0.776 | 0.953 | 0.909 | 0.941 | 0.911 | 0.883 | 0.875 |
| | P.I. | 0.911 | 0.784 | 0.957 | 0.843 | 0.965 | 0.970 | 0.770 | 0.915 | 0.947 | 0.968 | 0.905 | 0.757 | 0.855 | 0.806 | 0.719 | 0.841 |
| | U.N. | 0.919 | 0.872 | 0.989 | 0.942 | 0.983 | 0.989 | 0.625 | 0.771 | 0.989 | 0.984 | 0.900 | 0.755 | 0.901 | 0.902 | 0.552 | 0.883 |
| | P.N. | 0.817 | 0.953 | 0.816 | 0.960 | 0.955 | 0.887 | 0.969 | 0.994 | 0.927 | 0.873 | 0.950 | 0.908 | 0.961 | 0.958 | 0.784 | 0.909 |
| | O.A. | 0.859 | 0.886 | 0.841 | 0.928 | 0.959 | 0.908 | 0.826 | 0.933 | 0.930 | 0.903 | 0.926 | 0.826 | 0.916 | 0.905 | 0.739 | 0.880 |
| | **Kappa** | 0.718 | 0.756 | 0.586 | 0.816 | 0.907 | 0.779 | 0.633 | 0.824 | 0.776 | 0.787 | 0.853 | 0.654 | 0.826 | 0.785 | 0.450 | 0.754 |

Note: I.L., impervious landscape, CR, cropland-prevalent impervious landscape, BS, bare soil-prevalent impervious landscape, VG, vegetation-prevalent impervious landscape, P.I., producer's accuracy of impervious surfaces, U.I., user's accuracy of impervious surfaces, P.N., producer's accuracy of non-impervious surfaces, U.N., user's accuracy of non-impervious surfaces, O.A., overall accuracy.

To intuitively compare the performance of these six impervious products, five validation regions, including two bare soil-prevalent regions (Phoenix and Niamey), one vegetation-prevalent city (New York) and two cropland-prevalent regions (Winnipeg and Bangkok), were selected in Fig. 9. Specifically, in the first bare soil prevalent region of Phoenix, the NUACI-2015 obviously under-estimated the impervious surfaces in the center of Phoenix city. The causes of omission maybe came

from the threshold method used by the NUACI-2015. Liu et al. (2018) developed a novel NUACI index to enhance the impervious surfaces and suppressed the non-impervious surfaces and then found an optimal threshold for NUACI index to split the impervious and non-impervious surfaces. However, the NUACI values of rural villages and roads were usually located in the mixed areas of impervious and non-impervious surfaces, so the NUACI-2015 had great ability for large-size impervious surfaces but with poor performance for fragmented impervious surfaces. FROM_GLC-2015 performed well in the central city but missed impervious objects over peripheral urban. For example, the enlargement region (red rectangle), composited by sparse buildings and bare soils, was underestimated by the FROM_GLC-2015. This omission error maybe came from the sparse training samples (91,433 training samples in the globe) (Gong et al., 2013). The GHSL-2015, accurately capturing the central and peripheral impervious objects, had significant agreement with the MSMT-2015, it achieved the user's accuracy of 0.940 and producer's accuracy of 0.995 in this region (Table 2). As for the GlobeLand30-2010, there was little omission for the fragmented impervious objects over peripheral urban because of the temporal interval of 5 years and the minimum $4 \times 4$ mapping unit (Chen et al., 2015). The HBASE-2010 had biggest impervious areas among several global products but it misclassified the vegetation and bare soils into impervious surfaces in the urban central, so it had highest commission error of 9.5% in Table 2. As for the second bare soil prevalent city of Niamey, these products, except for the GHSL-2015 which had smaller impervious area than other products and missed the peripheral impervious objects, had similar performance with the Phoenix: the NUACI-2015 had high omission error especially for the fragmented objects, the HBASE-2010 lost the impervious details and achieved highest commission error of 5.3% in Table 2, the GlobeLand30-2010 missed some small objects (the limitation of minimum $4 \times 4$ mapping unit) and peripheral impervious objects caused by the temporal interval, and the FROM_GLC-2015 had great performance on the dense impervious areas but it was under-estimated over peripheral areas.

Next, in the vegetation-prevalent region of New York, six products generally had similar identification results and accurately captured the spatial distribution of New York city, so they achieved high mapping accuracy exceeding 90% in Table 2. However, from a detail perspective, there were still differences between these products. Specifically, NUACI-2015 performed well in the central of city but missed the sparse impervious objects over the peripheral city, for example, the enlargement region (red rectangle) illustrated the mixture of vegetation and sparse buildings over the peripheral city, the NUACI-2015 and GlobeLand30-2010 had smaller impervious areas than other products. The HBASE-2010 still suffered the highest commission error of 8.5% and had biggest impervious areas because it misclassified the bare soils and vegetation in the central city into impervious surfaces (blue rectangles). The GHSL-2015, FROM_GLC-2015 and MSMT-2015 achieved higher mapping accuracy because they captured both dense and sparse impervious objects in the central and peripheral city.

Lastly, in the two cropland-prevalent cities of Bangkok and Winnipeg, the MSMT-2015 had greater advantages and achieved highest user's accuracy of 95.1% and 100% compared to the NUACI-2015 of 69.5% and 77.7%, the FROM_GLC-2015 of 71.7% and 85.4%, the GHSL-2015 of 61.9% and 89.9%, the GlobeLand30-2010 of 31.0% and 74.9%, and HBASE-2010 of 80.1% and 91.1% in Table 2. Fig.9 intuitively illustrated the performance of each product. GlobeLand30-2010 had smaller impervious areas in the central city because of the temporal interval and missed the road networks due to the minimum mapping

unit of 4×4. As a result, the GlobeLand30-2010 achieved the lowest user's accuracy. NUACI-2015 captured impervious surfaces in the central city but missed the road networks and sparse village buildings in the peripheral cities. FROM_GLC-

590 2015 and HBASE-2015 had similar performance in these two regions, which captured medium and large cities but missed the road networks and villages buildings. As HBASE-2010 contained the OpenStreetMap data to provide information on major road network (Wang et al., 2017a), the omission error of the HBASE-2010 was relatively low and only these village roads and buildings were missed, however, it still suffered serious over-estimation problem. Especially in the Bangkok city, the non-impervious pixels (bare soils, water, and vegetation) was misclassified as impervious surfaces. Therefore, the HBASE-2010

595 reached the highest commission error among these impervious products in Table 2.

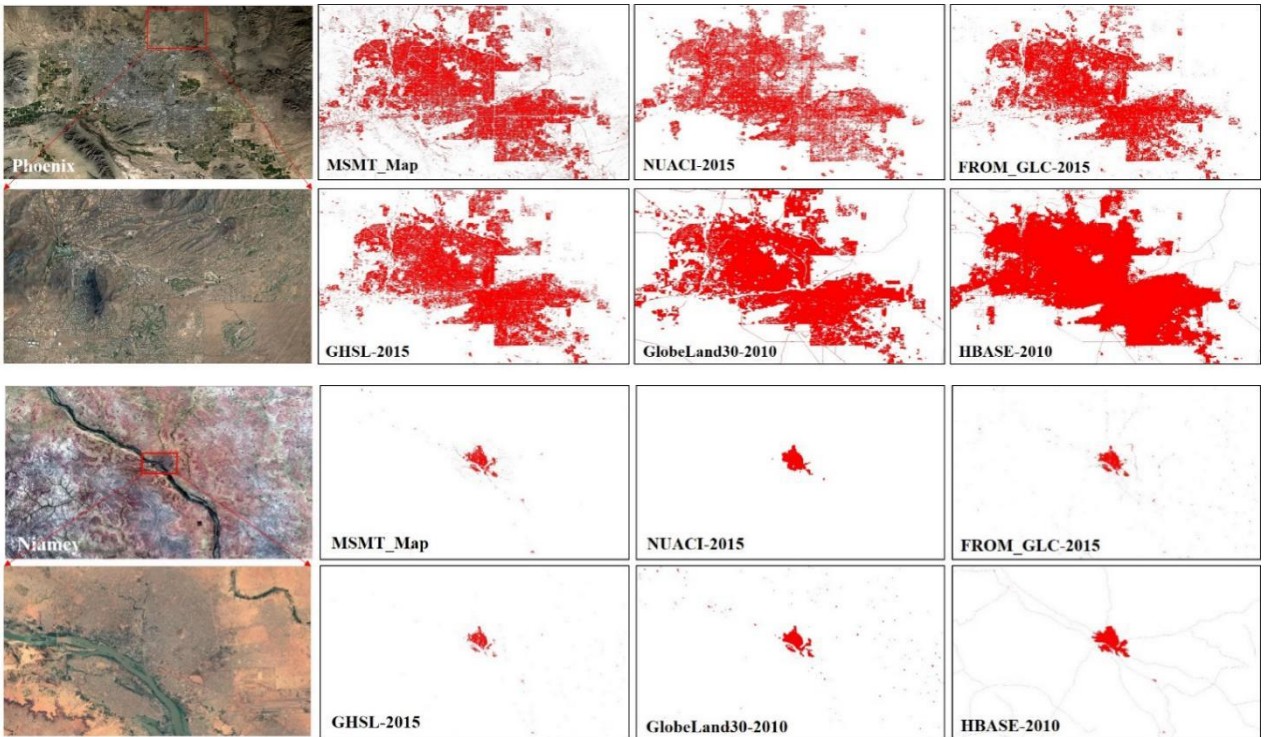

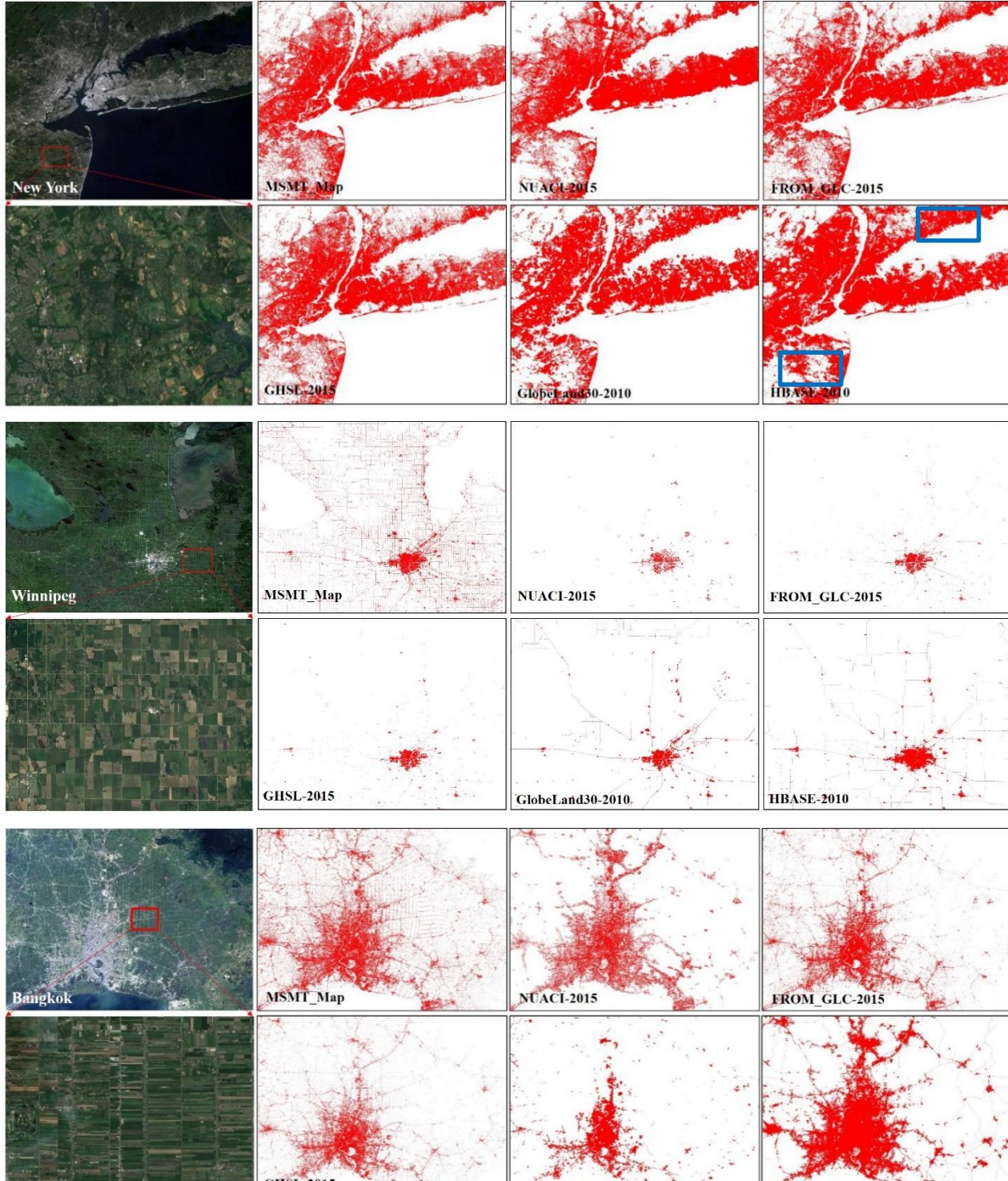

**Figure 9: Comparisons between the MSMT_RF-based impervious surface maps and other products (corresponded to the NUACI products developed by Liu et al. (2018), the FROM_GLC products developed by Gong et al. (2013), the GHSL products developed by Florczyk et al. (2019), the GlobeLand30 products developed by Chen et al. (2015), and the HBASE products developed by Wang et al. (2017a), respectively) for five regions with various impervious landscapes.**

## 6 Discussion

### 6.1 Reliability and sensitivity of the global training samples

In contrast to other classification-related studies that require manual efforts to collect training samples (Gao et al., 2012; Im et al., 2012; Zhang et al., 2016), we overcame the expensive cost of collecting accurate and sufficient training samples at a global scale. To ensure the accuracy and reliability of the training samples, a combination of the GlobeLand30-2010 land-cover product, which had been validated to have a producer's accuracy (which measures the commission error) of 94.7% for impervious surfaces (see Section 5.4), and DMSP-OLS NTL imagery was adopted to guarantee the reliability of each sample. As it was difficult and challenging to evaluate the accuracy of all the training samples, we randomly selected 1% of the total training samples (in Section 3) including 34,990 non-impervious and 9,840 impervious points to measure the reliability of the global training samples. After careful checking, we found that these training samples achieved accuracies of 91.9% and 99.5% for impervious and non-impervious surfaces, respectively.

Meanwhile, even if the training samples still contained a small number of erroneous points, the random forest model has been demonstrated to be resistant to noise and presence of erroneous samples (Belgiu and Drăguţ, 2016). In this study, we randomly changed the category of a certain percentage of the 34,990 samples and used the ''noisy'' samples to train the random forest classifier. Fig. 10 illustrates the overall accuracy and impervious producer's accuracy decreased for the increased percentage of erroneous samples. It was found that the overall and impervious producer's accuracy remained stable when the percentage of erroneous samples increased from 1% to 20% while it rapidly decreased when the percentage of erroneous samples was higher than 20%. Similarly, Gong et al. (2019) also found that the decrease in overall accuracy was less than 1% when the error in the training samples was less than 20%.

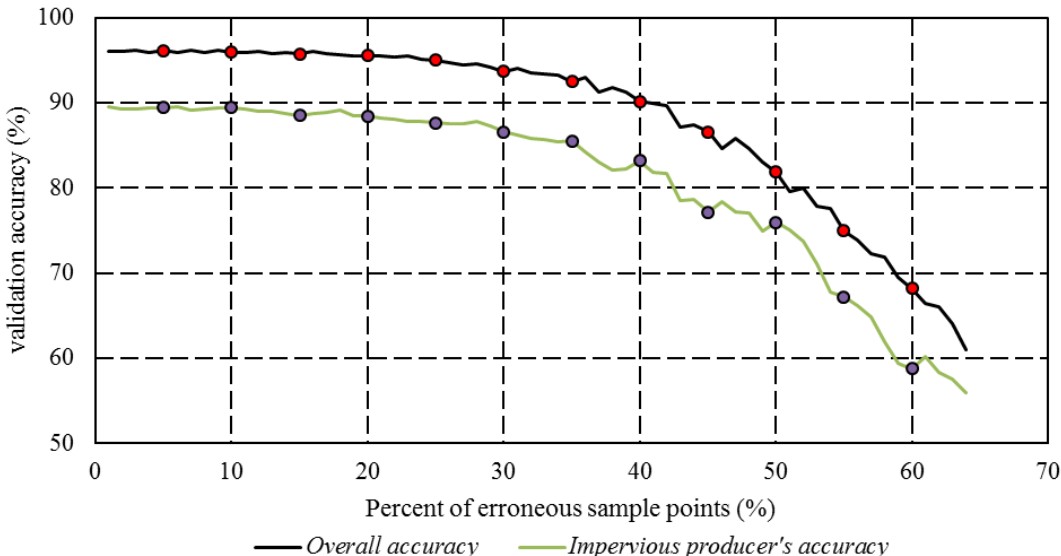

Figure 10: **Sensitivity analysis showing the relation between the classification accuracy and the percentage of erroneous samples points.**

Therefore, the reliability and sensitivity analysis indicated that: (1) the random forest model is resistant to noisy training samples and performs well if the percentage of erroneous samples is lower than 20%; and (2) the training samples derived from the GlobeLand30 and DMSP-OLS NTL imagery were accurate enough for use in global impervious surface mapping.

## 6.2 Limitations of the proposed method

Although the proposed MSMT_RF method has been demonstrated to have the ability to produce the accurate impervious surface products, there are still some limitations to the method. First, as the training samples derived from the GlobeLand30-2010 are restricted to a 9×9-pixel local window and further refined by the integration of MODIS EVI and VIIRS NTL imagery, low-density impervious samples might be omitted and cause further omission of low-density impervious surfaces (rural villages, small roads and so on). Although, in this study, spatially adjacent training samples from the surrounding 3×3 areas were imported to reduce the omission of low-density samples, according to the accuracy assessment, higher omission errors were found in low-density regions (Lhasa and Ntuman) than in high-density regions (New York and Moscow). Therefore, our future work will pay more attention to the omission of low-density impervious surfaces.

Secondly, as Weng (2012) pointed out, mixed pixels are common in medium-resolution imagery due to the limitations of the spatial resolution and spectral heterogeneity of the landscape. The effectiveness of 'hard' classifiers is easily affected by these mixed pixels (low-density impervious pixels also constitute mixed pixels). Due to the proportion of impervious surfaces within a pixel, impervious surface areas are often overestimated in urban areas or underestimated in rural areas when using medium-resolution images (Lu and Weng, 2006). Therefore, our future work will focus on simultaneously producing the likelihood

('soft' probability) of each pixel being an impervious surface. At present, some scientists have produced continuous impervious fractions at a regional scale: for example, Okujeni et al. (2018) used the support vector regression method to estimate the fraction of impervious surfaces at the pixel scale.

## 7 Data availability and user guidelines

The global impervious surface map data set generated in this paper is available on Zenodo:
https://doi.org/10.5281/zenodo.3505079 (Zhang and Liu, 2019).

To facilitate the readers to reproduce this work, Table 3 gives the details of the datasource and platform information of the datasets and processes in this study. The input remote sensing datasets and products came from three parts including: GEE platform, free access websites and our group. Specifically, five kinds of basic datasets in section 2.1 were available at GEE platform. The five impervious surface products in section 2.2 were downloaded from the free access websites from National
Geomatics Center of China, Tsinghua University, Sun Yat-sen University, National Aeronautics and Space Administration (NASA), and Joint Research Centre (JRC). The validation samples were produced by our group using visual interpretation.

Further, the process of derivation of global training samples was implemented by using the multi-source datasets at localhost computation platform, and the random forest classification at each $5°\times5°$ regional grid was developed by our group on the GEE platform using JavaScript language. The importance of multi-source and multi-temporal features and reliability and
sensitivity of global training samples were analyzed at the localhost Python computation environment.

**Table 3. The detailed information of the datasets and processes in this study**

|  | Datasource and platform | Detailed datasets and processing steps |
|---|---|---|
| Datasets | Google Earth Engine platform | Landsat-8 optical, Sentinel-1 SAR, VIIRS NTL, MODIS EVI and STRM/ASTER DEM topographical imagery |
|  | Free download websites | GlobeLand30-2010, FROM_GLC-2015, NUACI-2015, HBASE-2010 and GHSL-2015 products |
|  | Our Group | Validation samples |
| Processes | Google Earth Engine platform (JavaScript language) | The random forest classification at each $5°\times5°$ regional grid |
|  | Localhost platform (Python environment) | Derivation of global training samples<br>The importance of multi-source and multi-temporal features<br>reliability and sensitivity of global training samples |

## 8 Conclusions

Due to the spectral heterogeneity and complicated make-up of impervious surfaces, large-area impervious mapping is challenging and difficult. In this study, a global 30-m impervious surface map was developed by using multi-source, multi-
temporal remote sensing data based on the Google Earth Engine platform. First, the global training samples were automatically derived from the GlobeLand30-2010 land-cover product together with VIIRS NTL and MODIS EVI imagery. Then, a local

adaptive random forest model was trained using the training samples and multi-source and multi-temporal datasets for each 5 °×5 °geographical grid. Following that, the global impervious map produced by mosaicking a large number of 5 °×5 °regional impervious surface maps was validated by comparing it with several existing products (GlobeLand30-2010, FROM_GLC-2015, NUACI-2015, HBASE-2010 and GHSL-2015) using approximately 11,942 interpretation samples. The results indicated that the MSMT_RF-based impervious surface map achieved the highest overall accuracy 0.951 and kappa coefficient of 0.898 compared with 0.896 and 0.780 for FROM_GLC-2015, 0.856 and 0.695 for NUACI-2015, 0.903 and 0.794 for GHSL-2015, 0.884 and 0.753 for GlobeLand30-2010, and 0.880 and 0.754 for HBASE-2010 using all 15 regional validation data. Therefore, it can be concluded that the global 30-m impervious surface map produced by the proposed MSMT_RF method is accurate and reliable for use in global impervious surface mapping.

## Appendix: List of abbreviations and acronyms

**Table A1 List of abbreviations and acronyms**

| Abbreviation | Definition |
| --- | --- |
| DEM | Digital elevation model |
| DMSP OLS | Defense Meteorological Satellite Program Operational Linescan System |
| EANTLI | EVI-adjusted night-time light index |
| EVI | Enhanced vegetation index |
| FROM_GLC | Finer Resolution Observation and Monitoring of Global Land Cover |
| GEE | Google Earth Engine |
| GHSL | Global 30-m Human Settlement Layer |
| GlobeLand30 | 30 m Global Land Cover data product |
| HBASE | Human Built-up and Settlement Extent |
| MSMT_RF | Multi-source, multi-temporal random forest classification |
| NDBI | Normalized Difference Built-up Index |
| NDVI | Normalized Difference Vegetation Index |
| NDWI | Normalized Difference Water Index |
| NLCD | National Land Cover Dataset |
| NTL | Night-time light |
| NUACI | Normalized Urban Areas Composite Index |
| $R^2$ | Coefficient of determination |
| RMSE | Root mean square error |
| RF | Random forest |
| SAR | Synthetic aperture radar |
| VIIRS | Visible Infrared Imaging Radiometer |

**Author contributions.** Conceptualization, Liangyun Liu; Investigation, Xiao Zhang; Methodology, Liangyun Liu and Xiao Zhang; Software, Xiao Zhang and Xidong Chen; Validation, Xiao Zhang, Shuai Xie, Xidong Chen and Yuan Gao; Writing – original draft preparation, Xiao Zhang; writing—review and editing, Shuai Xie, Bing Zhang and Changshan Wu.

**Competing interests.** The authors declare that they have no conflict of interest.

**Financial support.** This research was funded by the Key Research Program of the Chinese Academy of Sciences (ZDRW-ZS-2019-2), the National Natural Science Foundation of China (41825002), and the Strategic Priority Research Program of the Chinese Academy of Sciences (XDA19080304).

**Acknowledgments.** We gratefully acknowledge the free access of GlobeLand30 land-cover products provided by National Geomatics Center of China, the FROM_GLC land-cover products provided by Tsinghua University, the NUACI impervious surface products provided by Professor Xiaoping Liu at Sun Yat-sen University, the GHSL impervious surface products produced by National Aeronautics and Space Administration, and HBASE produced by the Joint Research Centre.

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
