# Peer review of "Development of a global 30-m impervious surface map using multisource and multi-temporal remote sensing datasets with the Google Earth Engine platform"

_Earth System Science Data, 2019_

## Referee Comment (RC1) · Anonymous Referee #1 · 3 Mar 2020

This manuscript introduced a global scale impervious surface map generated with multi-source remote sensing datasets, and comparative analysis suggested that the developed map outperformed the state-of-art land cover products. Despite producing a global impervious surface map using manifold datasets is an important contribution to the global land cover dataset, a major revision suggestion may be given from my side.

Major revision points are as follow:

[Figure]

1. The review of impervious surface datasets should be further improved. Here I recommend a few examples: Global Man-made Impervious Surface (GMIS) Dataset, Copernicus land monitoring surface – high resolution layer imperviousness (although this dataset only covers Europe continent, it can be used as training and validation sample source), NLCD imperviousness products, Global Human Built-up And Settlement Extent (HBASE) Dataset.

2. The scientific importance of your dataset should be enhanced. Demonstration of multiple dataset contributions to land cover classification was not straightforward. Data (spectral or radar) characteristics on different land covers (e.g. vegetation, impervious surface and bare soil) should be revealed in detail.

3. The accuracy assessment experiment should be improved and expanded to multiple urban landscape types (e.g. globally selecting validation sites in more bare soil prevalent cities and vegetation prevalent cities), so that readers can clearly understand how multiple datasets work in land cover mapping under varying landscape conditions (the same reason as the 2nd comment).

4. The training sample source/method may not be scientifically sound. GlobeLand30 was adopted as impervious surface training sample source, however, this global land cover product provides users with artificial layer but not impervious surface layer. The "impervious surface land cover" used in this study is actually a mixture land cover of vegetation, impervious surface and bare soil in urban area.

5. More explanations may be required to arguments in this paper.

Here I am left with a limited number of questions about the specifics of implementation and implications of the method and results as well as clarity of this manuscript, which I note below.

[Questions for "introduction"]

"However, Gao et al. (2012) explained that these coarse-resolution global impervious

surface maps were not suitable for many applications and policy makers at local or regional scales (Line 35)." This part is not understandable, why the previous impervious surface maps are not suitable for certain applications and policy makers? Could you please explain it with more straightforward instances?

"(Chen et al., 2015; Gao et al., 2012; Goldblatt et al., 2018; Gong et al., 2019; Gong et al., 2013; Homer et al., 2015; Li et al., 2018; Liu et al., 2018; Sun et al., 2017). Line 50". Because of their similar works as you did in this paper (i.e. global or regional impervious surface maps), it is necessary to give more introduction to previous land cover datasets, and to present the importance of your work.

"However, these land-cover products focus on the overall accuracy of the mapping of all land-cover types rather than that of impervious surfaces alone (Line 55)". It may be confusing here. It implies the land cover products that focus the overall accuracy deliver low quality impervious surface map. What difference existing between "focusing on overall accuracy" and "focusing on impervious surface alone"?

"However, the NUACI product had a relatively poor performance in terms of producer's accuracy (0.50–0.60) and user's accuracy (0.49-0.61). Therefore, an accurate impervious surface map at fine spatial resolution is still urgently needed (Line 55)". Why did you only mention accuracy of NUACI here? How about other land cover datasets?

"However, these spectral mixture methods can produce underestimates in areas whether the density of impervious surfaces is high and overestimates in areas of low density (Sun et al., 2017; Weng, 2012) (Line70)". Spectral unmixing technique may have underestimate and overestimate issues, but how about its overall or average accuracy when comparing it with pixel-level mapping approaches?

[Technical questions for land cover mapping process]

The "data preprocessing" and "mapping approach" was mixed up, which makes readers difficult to capture the point of datasets and classification methods, so I may suggest

splitting them into different sections.

Questions on remote sensing datasets for classification (Lines 120). Descriptions of purpose and necessity for different remote sensing datasets were not clear, better wording in "Datasets" may be required.

Do five data sources contribute equally to classification? How do they theoretically work for differentiating different land covers? (Line 120)

How does C-band SAR imagery contribute to differentiate impervious and pervious surfaces? How do artificial buildings, forests, grassland, and bare soil respond to SAR imagery? Please clarify it. (Line 130)

How do EVI imagery work for classification procedure? (Line 145) Why do you involve DEM dataset in land cover mapping? How does this dataset work in differentiating impervious and pervious surfaces? (Line 150)

Questions on introducing state-of-art global impervious surface products. (Lines 205). GlobeLand30 actually does not provide impervious surface land cover, please adopt other global land cover dataset instead (Line 160). Furthermore, the review of the published global impervious surface datasets should be improved. For instance, three important global (or continental) impervious surface datasets – global man-made impervious surface (GMIS) dataset, NLCD impervious surface layer and global human built-up and settlement extent (HBASE) dataset- were not introduced.

Questions on selecting training samples (Lines 205). The impervious surface training samples were selected based on GlobeLand30 map. However, GlobeLand30 only provides "artificial surface" which consists of impervious surfaces and small patch vegetation areas in urban area. Thus, the training samples of MSMT_RF could be no longer reliable although extra datasets were used for samples filtering.

The training samples for classifier may be collected from other impervious surface datasets instead of GlobeLand30 map.

It is not clear that how and why twelve sampling sites (i.e. high-density sites, medium-density sites and low-density regions) were selected? How do spectral features vary among these sites? What features were exhibited by different density regions? This information should be updated.

Moreover, data preprocessing procedures and mapping methods are mixed up, which makes manuscript confusing. I suggest separating them into two sections. In particular, more explanations of the preprocessing operations should be made,instead of only citing reference literature.

Some parts of introduction to data preprocessing were also not understandable. Here is an example: "the suburban areas or rural villages were also easy to confused with croplands (Li et al., 2015)". It is not reasonable to compare a land use element (suburban area) with a land cover element (cropland). Explanations are always required for each of your arguments.

Questions on accuracy assessment (Lines 315). Two accuracy assessment was conducted respectively in "fraction" way and "classified pixel" way. How much difference do the two accuracy assessment methods make? What special information can be provided by each method?

Questions on Figure 5. As mentioned in previous questions, further review of currently available global impervious surface maps is needed. In the revised manuscript, I suggest adding error bars for progressive fraction intervals (e.g. 0.05, 0.1, 0.15, 0.2, ..., 1.0).

Confusing parts in Section 4.2. "As the stratified random sampling strategy was applied to each validation region independently, the low and medium density regions were easier to select these mixed impervious validation points (simultaneously containing the impervious and non-impervious surfaces in the 30-m ×30-m validation window and the impervious areas exceed the predefined threshold of 50%) which were most difficult to identify for impervious surface mapping (Line 380)." What information did you want

to present?

Questions for Figure 6. It is clear to show difference between impervious surface maps but not clear to visually compare RGB pixels with your maps. The RGB satellite images of macro areas may not be suitable to compare it with classified land cover map. Subset urban areas are preferred so that readers can clearly see how well the map is classified. Besides, I may not agree that "low, medium, high- density" areas are representative for comparison. To improve the figure, I suggest globally selecting urban areas with different landscapes (e.g. desert landscape urban areas such as Phoenix city, vegetation prevalent cities such as New York City). Futhermore, please do more works in reviewing global impervious surface datasets.

[Questions for "Discussion"]

Figure 7 is an experiment result, and it should be moved to "Result".

"The importance of all 37 training features for the six regions is illustrated in Fig. 7. These results indicate that the Sentinel-1 440 SAR features (VV and VH) had the greatest contribution to the final decision in most regions because SAR images can provide information about the structure and dielectric properties of the surface materials (Line 440)". What VV and VH feature difference is revealed between different land covers (e.g. impervious surface, forest, croplands, bare soil, water)?

"Similarly, Zhu et al. (2012) demonstrated that the inclusion of multi-temporal imagery increased the accuracy by 8.9%. Schug et al. (2018) also found that bi-seasonal information could produce a more reliable performance than a single-year composited image. Therefore, temporal variability can be considered an important addition to accurate impervious surface mapping (Line 455)." Discussion and explanation should be made. Please exactly explain the theory in which how these datasets work for improving classification accuracies. Which land cover accuracy is improved by including these datasets?

"Similarly, Clarke et al. (1997) explained that topographical variables (slope, aspect and DEM) contribute a lot to impervious surface mapping. These features are, therefore, indispensable in the accurate mapping of impervious surfaces in complex landscapes (Line 465)." Clarke et al. (1997) was cited without further explanation. Readers would like to know mechanism of topographical variable contributing to impervious surface mapping?

---

## Short Comment (SC2) · 25 Mar 2020

This paper presents a new global 30m impervious surface map produced with multi-source and multi-temporal remote sensing datasets and random forest (MSMT_RF). Compared with the currently available impervious products (i.e., GlobeLand30, FROM_GLC and NUACI), this MSMT_RF-based product has higher overall accuracy and kappa coefficient, which are 96.6% and 0.90, respectively. The superiority of the MSMT_RF-based product stems from two significant innovations of the method proposed in this study. First, multi-source and multi-temporal remote sensing data are

25

combined to produce the impervious surface map. The comprehensive information provided by the combined data is useful in classifying land cover types, so the superiority of the MSMT_RF-based product in comparison with the other products is convincing. Second, a novelty method is proposed for selecting training samples based on the available impervious product and VIIRS NTL and MODIS EVI imagery. This method allows for the fully automatic selection of training samples to avoid manual training sample selection, which is time-consuming and laborious, especially at a global scale. This method has significant implications for producing more perfect global data products based on existing data products. I believe this study is a breakthrough over previous works in impervious surface mapping and will appeal to a broad readership. However, there are still some minor issues that should be addressed before final publication.

Line 35, "urban the environment" should be "urban environment"

Figure 1, I cannot see the blue rectangles but only black points, which are supposed to be the blue rectangles. The authors should figure out how to make blue rectangles clear.

Why did the authors select training samples based on Globe30 product but not FORM_GLC, which is also a 2015 product and seems to be more appropriate? Please elaborate.

Figure 5, please provide the label of axes.

Table 2. How the different categories, e.g., high, low, medium, are defined? Are they defined quantitatively or subjectively? Please elaborate.

Figure 6. I suggest the authors to provide the location information (e.g., city name or latitude-longitude grid) of these areas. It will allow readers to check ground truth in Google Earth.

Page 20, the authors found that the importance of Landsat textural features is low, whereas previous studies confirmed the contribution of textural features to impervi-

ous mapping. More explanations can be given on this contradiction. One possible explanation may be the different data sets. Many studies have indicated that textural information is helpful in land cover classification, especially in high-resolution images. Shaban and Dikshit (2001) used the textural information in SPOT images, while the authors used that in Landsat-8 images. The difference in spatial resolution may cause the different contribution of textural features in impervious surface mapping.

Page 20, I agree with that the improvement made by this study is mainly due to the combination of the multi-source and multi-temporal information, but it may be misleading to state that the classification-based method performed better than spectral index-based method since they are performed based on the different data sets. I do not think the classification-based method can achieve a high accuracy only with Landsat data.

---

## Referee Comment (RC2) · Anonymous Referee #2 · 11 Apr 2020

The paper describes a new impervious surface dataset developed by combining several remote sensing instrumentation at 30m resolution. As described in the introduction, several datasets describing impervious studies exist at a global scale. The strength of this paper is in my opinion the use of multi-sensor information and the use of an open-source platform the generate these maps (Google Earth Engine). Furthermore, a relatively good accuracy of the map is achieved compared to three other impervious surface products. The paper is very well written and is easy to follow. The introduction also gives a very good overview of current existing literature. The paper is

very mature and contains all information one would expect for this kind of work. Most of the comments that popped in my mind while reading the paper were assessed later in the manuscript. As such, for me only minor revisions are necessary. I describe some comments below.

General comments

- Training points are achieved from Globeland30 and are not independent based on independent experts (which is done for the validation data). Several checks are done on the training data, but you are still using a derived product with errors to train your model. In the discussion, this problem is assessed (section 5.2). However, I would state this more clear that the training sample can contain errors in the material and methods section and potentially move the discussion to the material and methods section or refer in the material and methods section that this problem will be assessed later.

- Only homogeneous training points from Globeland30 are included. Therefore, the training points are always clear impervious surfaces leading to only clear impervious surfaces to be classified later. Don't you underestimate the total amount of impervious surfaces then in your final product? How does the total % of impervious surface compare to Globeland30, GLC and NAUCI for the globe? This can maybe be compared to the results presented in figure 5

- The validation points are retrieved from 12 regions. How representative are these regions for the globe? Since impervious areas might have very different characteristics depending on the region. For Africa for example, the validation points are achieved for two big cities only.

Specific comments

- Globeland30 data from 2010 is used as training data. How do you account for changes in urban areas between 2010-2015? You state that there is an irreversible state from non-impervious to impervious surfaces, but this could mean that some im-

pervious surfaces in 2010 have now changed to impervious in 2015.

- Add to table 1 that the 15 + 85 percentile are used for the Landsat bands and vegetation indices

- Line 265, remove 'the'

---

## Author Comment (AC1) · 2 May 2020

**Response to comments**

**Paper #:** essd-2019-200
**Title:** Development of a global 30-m impervious surface map using multi-source and multi-temporal remote sensing datasets with the Google Earth Engine platform
**Journal**: Earth System Science Data

**Reviewer #1**

This manuscript introduced a global scale impervious surface map generated with multi-source remote sensing datasets, and comparative analysis suggested that the developed map outperformed the state-of-art land cover products. Despite producing a global impervious surface map using manifold datasets is an important contribution to the global land cover dataset, a major revision suggestion may be given from my side.

Great thanks for the comments. The manuscript has been improved according to your and other reviewers' comments.

1. The review of impervious surface datasets should be further improved. Here I recommend a few examples: Global Man-made Impervious Surface (GMIS) Dataset, Copernicus land monitoring surface – high resolution layer imperviousness (although this dataset only covers Europe continent, it can be used as training and validation sample source), NLCD imperviousness products, Global Human Built-up And Settlement Extent (HBASE) Dataset.

Thanks for the comment. To make the cross-comparisons more comprehensive, two global impervious products (HBASE and GHSL) have been added in the Section 2 Datasets as (as GMIS and HBASE were companion datasets and GMIS provided continual imperviousness products, only HBASE was included):

[revised manuscript text omitted]

4. The training sample source/method may not be scientifically sound. GlobeLand30 was adopted as impervious surface training sample source, however, this global land cover product provides users with artificial layer but not impervious surface layer. The "impervious surface land cover" used in this study is actually a mixture land cover of vegetation, impervious surface and bare soil in urban area.

Great thanks for the comment. The reasons why we used the GlobeLand30 to derive the global training samples are included as follows.:

1) After carefully checking, there was great consistency for the definition of impervious surface between GlobeLand30 and our study. Specifically, the GlobeLand30 in (Chen et.al 2015) defined "Artificial surfaces mainly consists of urban areas, roads, rural cottages and mines, which are primarily based on asphalts, concrete, sand and stone, bricks, glasses, and other materials";

-our study: "Impervious surfaces are usually covered by anthropogenic materials which prevent water penetrating into the soil (Weng, 2012), which are primarily composited by asphalts, sand and stone, concrete, bricks, glasses, etc."

Similarly, the NUACI products also shared same definition with GlobeLand30 as "the term 'urban land' in this paper refers to 'impervious surface', i.e., artificial cover and structures such as pavement, concrete, brick, stone and other man-made impenetrable cover types (Liu et al. 2018)" .

2) The GlobeLand30 had several advantages over other impervious products (NUACI, FROM_GLC and GHSL..) including: it was developed by combining pixel-based classification, multi-scale segmentation and manual editing based on high-resolution imagery, so almost impervious objects in GlobeLand30 were checked by visual interpretation. In addition, the non-impervious training samples in this study included three sub-classes (cropland, bare soil and other non-impervious land-cover types), if we chose the NUACI or GHSL products, these non-impervious samples similar to impervious surface cannot be completely collected. The reasons have been added in the Section 3 - "Collection of global training samples" as:

"The GlobeLand30 land-cover product was used to derive global training samples because it had many advantages including: (1) the impervious surface layer in GlobeLand30 was accurately developed by combining the pixel-based classification, multi-scale segmentation and manual editing based on high resolution imagery and validated to achieve an user's accuracy of 86.7%; (2) it simultaneously contained the impervious surface and other land-cover types similar to impervious surface (such as cropland and bare land), so the global training samples including several non-impervious land-cover types could be easily collected to build the RF model for accurately mapping of impervious surface. "

5. More explanations may be required to arguments in this paper.
Here I am left with a limited number of questions about the specifics of implementation and implications of the method and results as well as clarity of this manuscript, which I note below.
Great thanks for the comment. These questions have been answered one by one in the following comments.

"However, Gao et al. (2012) explained that these coarse-resolution global impervious surface maps were not suitable for many applications and policy makers at local or regional scales (Line 35)." This part is not understandable, why the previous impervious surface maps are not suitable for certain applications and policy makers? Could you please explain it with more straightforward instances?
Thanks for the comment. To make the expression more straightforward, this sentence has been rewritten as:
"However, since the complex characteristics of impervious landscapes and inherent resolution of human activity, coarse-resolution global impervious surface maps were not suitable for many

applications and policy makers at local or regional scales, for example, the urban-rural pattern planning and road network monitoring usually required the fine spatial resolution impervious surface products (Gao et al., 2012)"

"(Chen et al., 2015; Gao et al., 2012; Goldblatt et al., 2018; Gong et al., 2019; Gong et al., 2013; Homer et al., 2015; Li et al., 2018; Liu et al., 2018; Sun et al., 2017). Line 50". Because of their similar works as you did in this paper (i.e. global or regional impervious surface maps), it is necessary to give more introduction to previous land cover datasets, and to present the importance of your work.

Thanks for the comment. Yes, it is necessary to give more details to previous land-cover datasets, this paragraph has been added as:

Specifically, the National Land Cover Dataset (NLCD) produced the first 30-m map of the United States including impervious surface as three separate land-cover types (Developed low, Developed medium, and Developed high intensity) using Landsat imagery, DMSP OLS and USGS National Elevation Dataset (NED) digital elevation data, and achieving the user' accuracy of 0.48~0.66 (Homer et al., 2004). Similarly, the FROM_GLC produced the global 30-m impervious surface map as an independent land cover type with the user' accuracy of 0.307 (Gong et al., 2013); GlobeLand30 combined the pixel-based classification, segmentation and manual editing based on the high resolution imagery to develop the 30-m impervious surface map as an independent layer with the user's accuracy of 0.867 (Chen et al., 2015). However, as sparse training samples of impervious surfaces cannot capture all relevant spectral heterogeneity when producing these land-cover products, the impervious surface layers usually suffered low accuracy except for GlobeLand30 (which includes manual interpretation). Therefore, a few studies proposed to independently produce the impervious surface products. For example, Liu et al. (2018) proposed the Normalized Urban Areas Composite Index (NUACI) method to produce a global 30-m impervious surface map and achieved an overall accuracy of 0.81-0.84 and a kappa values of 0.43-0.50. However, the NUACI product had a relatively poor performance in terms of producer's accuracy (0.50–0.60) and user's accuracy (0.49-0.61). Brown de Colstoun et al. (2017) combined the object-based segmentation, random forest classification and post-processing to develop the Global 30-m Man-made Impervious Surface (GMIS) and Human Built-up and Settlement Extent (HBASE) dataset in 2010 which achieved a kappa coefficient of 0.91 using scene-level cross validation in Europe (Wang et al., 2017b). Pesaresi et al. (2016) used the multi-temporal Landsat imagery and symbolic machine learning method to produce the Global 30-m Human Settlement Layer (GHSL) in 2014, and achieved a total accuracy of 96.28% and kappa coefficient of 0.3233 based on Land Use/Cover Area frame Survey (LUCAS) reference data. Therefore, an accurate impervious surface map at fine spatial resolution is still urgently needed using an efficient mapping method."-

"However, these land-cover products focus on the overall accuracy of the mapping of all land-cover types rather than that of impervious surfaces alone (Line 55)". It may be confusing here. It implies the land cover products that focus the overall accuracy deliver low quality impervious surface map. What difference existing between "focusing on overall accuracy" and "focusing on impervious surface alone"?

Thanks for the comment. This sentence has been rewritten as:

"However, as sparse training samples of impervious surfaces cannot capture all relevant spectral heterogeneity when producing these land-cover products, the impervious surface layers usually suffered

low accuracy except for GlobeLand30 (which includes manual interpretation). Therefore, a few studies proposed to independently produce the impervious surface products. For example...”

“However, the NUACI product had a relatively poor performance in terms of producer's accuracy (0.50–0.60) and user's accuracy (0.49-0.61). Therefore, an accurate impervious surface map at fine spatial resolution is still urgently needed (Line 55)”. Why did you only mention accuracy of NUACI here? How about other land cover datasets?

Thanks for the comment. The accuracies of other datasets have been added as:

the National Land Cover Dataset (NLCD) produced the first 30-m map of the United States including impervious surface as three separate land-cover types (Developed low, Developed medium, and Developed high intensity) using Landsat imagery, DMSP OLS and USGS National Elevation Dataset (NED) digital elevation data, and achieving the user' accuracy of 0.48~0.66 (Homer et al., 2004). Similarly, the FROM_GLC produced the global 30-m impervious surface map as an independent land cover type with the user' accuracy of 0.307 (Gong et al., 2013); GlobeLand30 combined the pixel-based classification, segmentation and manual editing based on the high resolution imagery to develop the 30-m impervious surface map as an independent layer with the user's accuracy of 0.867 (Chen et al., 2015).

Liu et al. (2018) proposed the Normalized Urban Areas Composite Index (NUACI) method to produce a global 30-m impervious surface map and achieved an overall accuracy of 0.81-0.84 and a kappa values of 0.43-0.50. However, the NUACI product had a relatively poor performance in terms of producer's accuracy (0.50–0.60) and user's accuracy (0.49-0.61). Brown de Colstoun et al. (2017) combined the object-based segmentation, random forest classification and post-processing to develop the Global 30-m Man-made Impervious Surface (GMIS) and Human Built-up and Settlement Extent (HBASE) dataset in 2010 which achieved a kappa coefficient of 0.91 using scene-level cross validation in Europe (Wang et al., 2017b). Pesaresi et al. (2016) used the multi-temporal Landsat imagery and symbolic machine learning method to produce the Global 30-m Human Settlement Layer (GHSL) in 2014, and achieved a total accuracy of 96.28% and kappa coefficient of 0.3233 based on Land Use/Cover Area frame Survey (LUCAS) reference data (Pesaresi et al., 2016).

“However, these spectral mixture methods can produce underestimates in areas whether the density of impervious surfaces is high and overestimates in areas of low density (Sun et al., 2017; Weng, 2012) (Line70)”. Spectral unmixing technique may have underestimate and overestimate issues, but how about its overall or average accuracy when comparing it with pixel-level mapping approaches?

Great thanks for the comment. The spectral unmixing techniques and pixel-level mapping methods represented the 'soft' and 'hard' classifications respectively, and had different advantages for impervious mapping. The accuracies of spectral unmixing techniques and pixel-level classification mainly depended on the reliability of endmember and training data respectively, so it was difficult to directly compare the performance of two methods. However, the spectral unmixing techniques had great difficulties to identify one suitable endmember to represent all types of impervious surfaces, so the pixel-level mapping approaches were more popular for impervious surface mapping. The disadvantages of spectral unmixing techniques have been added as:

“However, these spectral mixture methods can produce underestimates in areas with high density

impervious surfaces and overestimates in areas with low density impervious surfaces, **and may have great difficulties to identify one suitable endmember to represent all types of impervious surfaces** (Sun et al., 2017; Weng, 2012)"

The "data preprocessing" and "mapping approach" was mixed up, which makes readers difficult to capture the point of datasets and classification methods, so I may suggest splitting them into different sections.

Great thanks for the comment. Based on the suggestion, we split the data preprocessing of "Collection of global training samples" as an independent section 3.

Questions on remote sensing datasets for classification (Lines 120). Descriptions of purpose and necessity for different remote sensing datasets were not clear, better wording in "Datasets" may be required

Great thanks for the comment. Based on the suggestion, the descriptions of the remote sensing datasets have been added, the specific changes have been listed in the following comments.

Do five data sources contribute equally to classification? How do they theoretically work for differentiating different land covers? (Line 120)

Thanks for the comment. The functions of five datasets have been added as:

"In this study, three kinds of data sources including Landsat-8 optical imagery, Sentinel-1 SAR data and STRM/ASTER DEM topographical variables were selected and collected for the mapping of impervious surfaces across the world on the GEE platform. Furthermore, the combination of VIIRS NTL imagery and MODIS EVI products was used to derive the set of global impervious surface and non-impervious surfaces training data."

How does C-band SAR imagery contribute to differentiate impervious and pervious surfaces? How do artificial buildings, forests, grassland, and bare soil respond to SAR imagery? Please clarify it. (Line 130)

Thanks for the comment. The explanations why we imported the Sentinel-1 SAR imagery are added as:

"The Sentinel-1 satellite provides C-band SAR imagery at a variety of polarizations and resolutions. (Berger et al., 2012; ESA, 2016; Torres et al., 2012). **Due to the high dielectric properties of the building materials, the unique geometry of manmade features, and the special radar echo properties of artificial structures, the impervious surfaces usually had stronger backscattered signals than other land-cover types (such as: barren land, cropland and so on) in the SAR imagery.** In this study..."

How do EVI imagery work for classification procedure? (Line 145) Why do you involve DEM dataset in land cover mapping? How does this dataset work in differentiating impervious and pervious surfaces? (Line 150)

Thanks for the comment. The work of EVI imagery has been added as:

"The MODIS EVI imagery (MYD13Q1) from the MODIS V6 products contains the best available EVI data from among all the acquisitions obtained over a 16-day compositing period and has a spatial resolution of 250-m (Didan et al., 2015), **which was used to mitigate the NTL data's saturation**

**problem and exclude false positive impervious samples (vegetated samples in the urban) when deriving the global training samples**. In this study, ..."

The reasons why we import the DEM dataset in impervious surface mapping are added as:

"The Shuttle Radar Topography Mission digital elevation model (SRTM DEM), provided by the NASA JPL at a resolution of 1 arc-second (approximately 30 m) and covering the area between 60° north and 56° south (Farr et al., 2007), **was an useful auxiliary dataset for impervious surface mapping over mountainous areas because impervious surfaces mainly located in the flat areas and Sentinel-1 SAR data usually reflected high backscatter similar to the impervious surfaces in mountainous areas** (Ban et al., 2015)."

Questions on introducing state-of-art global impervious surface products. (Lines 205). GlobeLand30 actually does not provide impervious surface land cover, please adopt other global land cover dataset instead (Line 160). Furthermore, the review of the published global impervious surface datasets should be improved. For instance, three important global (or continental) impervious surface datasets – global man-made impervious surface (GMIS) dataset, NLCD impervious surface layer and global human built-up and settlement extent (HBASE) dataset- were not introduced.

Many thanks for the comment. After carefully checking the impervious surface definition of GlobeLand30 in Chen et al. (2015), we found there is consistency between GlobeLand30 and our study for defining impervious surface:

-GlobeLand30: "Artificial surfaces mainly consists of urban areas, roads, rural cottages and mines, which are primarily based on asphalts, concrete, sand and stone, bricks, glasses, and other materials";

-our study: "Impervious surfaces are usually covered by anthropogenic materials which prevent water penetrating into the soil (Weng, 2012), which are primarily composited by asphalts, sand and stone, concrete, bricks, glasses, etc. (Chen et al., 2015)."

Similarly, the NUACI products also shared same definition with GlobeLand30 as "the term 'urban land' in this paper refers to 'impervious surface', i.e., artificial cover and structures such as pavement, concrete, brick, stone and other man-made impenetrable cover types" (Liu et al. 2018)

Next, in order to improve the review of the published global impervious surfaces datasets, the Human Built-up and Settlement Extent (HBASE) and Global Human Settlement Layer (GHSL), have been added as: (Note: as HBASE and GMIS were companion datasets and GMIS was continues impervious fraction, we only selected the HBASE dataset):

The HBASE (Human Built-up and Settlement Extent) dataset was the first global 30-m dataset of man-made impervious cover derived from the Global Land Survey (GLS) Landsat data for 2010 (HBASE-2010) (https://sedac.ciesin.columbia.edu/data/set/ulandsat-hbase-v1). It was produced by combining meter-resolution training data (exceeding 20 millions), Open Street Map, VIIRS NTL, GLS Landsat SR and MODIS NDVI products, and achieved a kappa coefficient of 0.91 using scene-level cross validation in Europe (Wang et al., 2017a; Wang et al., 2017b).

The GHSL (Global Human Settlement Layer), a global information baseline describing the spatial evolution of the human settlements in the past 40 years, was developed by using symbolic machine learning model trained by the collected high-resolution samples, multi-temporal Landsat imagery in the epochs 1975, 1990, 2000, and 2015 (Florczyk et al., 2019). In this study, the GHSL impervious surface

map at 30-m for 2015 (GHSL-2015) (https://ghsl.jrc.ec.europa.eu/download.php) was employed for comparison analysis, which achieved an overall accuracy of 96.28% and kappa coefficient of 0.3233 validated using Land Use/Cover Area frame Survey (LUCAS) reference data (Pesaresi et al., 2016).

Questions on selecting training samples (Lines 205). The impervious surface training samples were selected based on GlobeLand30 map. However, GlobeLand30 only provides "artificial surface" which consists of impervious surfaces and small patch vegetation areas in urban area. Thus, the training samples of MSMT_RF could be no longer reliable although extra datasets were used for samples filtering. The training samples for classifier may be collected from other impervious surface datasets instead of GlobeLand30 map.

Great thanks for comment. The "artificial surface" in GlobeLand30 was defined as:"Artificial surfaces mainly consists of urban areas, roads, rural cottages and mines, which are primarily based on asphalts, concrete, sand and stone, bricks, glasses, and other materials", which is same as our definition for impervious surface: "Impervious surfaces are usually covered by anthropogenic materials which prevent water penetrating into the soil (Weng, 2012), which are primarily composited by asphalts, sand and stone, concrete, bricks, glasses, etc. (Chen et al., 2015)."

Next, the reasons why we chose the GlobeLand30 instead of other products (GMIS, GHSL, FROM_GLC and so on) were: (1) GlobeLand30 had the user' accuracy of 0.867 for impervious surface, and each impervious surface object was edited by manual interpretation, which greatly guaranteed the high confidence of impervious surface; (2) The training samples in this study contained impervious surface and non-impervious surfaces (barren land, cropland and other land-cover types), if we chose the GMIS or GHSL products, we cannot collect the training data of some non-impervious surfaces (barren land and cropland) which usually shared similar spectra with impervious surface. The reasons have been added in the Section 3 –"Collection of global training samples" as:

The GlobeLand30 land-cover product was used to derive global training samples because it had many advantages including: (1) the impervious surface layer in GlobeLand30 was accurately developed by combining the pixel-based classification, multi-scale segmentation and manual editing based on high resolution imagery and validated to achieve an user's accuracy of 86.7%; (2) it simultaneously contained the impervious surface and other land-cover types similar to impervious surface (such as cropland and bare land), so the global training samples including several non-impervious land-cover types could be easily collected to build the RF model for accurately mapping of impervious surface.

Finally, in order to guarantee the confidence of the training samples, we took two steps: (1) selecting the homogeneous areas as the candidate set; (2) using the EANTLI index to minimize the effects of classification error and the land-cover changes caused by the temporal interval.

It is not clear that how and why twelve sampling sites (i.e. high-density sites, medium density sites and low-density regions) were selected? How do spectral features vary among these sites? What features were exhibited by different density regions? This information should be updated.

Great thanks for the comment. The twelve sampling sites were randomly selected by combining the histogram of impervious fraction. However, these sites cannot demonstrate the importance of multi-source datasets. Therefore, based on the previous and latter suggestions, the sampling sites have been re-selected by combining the land-cover types.

To quantitatively assess the performance of the global impervious surface datasets, fifteen validation regions, covering different continents and various urban landscapes (the bare soil prevalent cities: Phoenix (PNX), Madrid (MDR), Riyadh (RYH), Niamey (NIM), Johannesburg (JHB), Ntuman (NTU) and Lhasa (LHS), vegetation prevalent cities: New York (NYK), Manaus (MNS), Moscow (MSC), San Paulo (SPL) and Melbourne (MBN), as well as cropland prevalent cities: Winnipeg (WIP), Bangkok (BGK) and Xi'an (XAN)), were selected (Fig. 1)

[Figure]

**Figure 1: The spatial distribution of the fifteen validation regions (blue) corresponding to regions of different impervious landscapes on different continents, together with the six 5°×5° validation regions (red) used to measure the variable importance.**

Moreover, data preprocessing procedures and mapping methods are mixed up, which makes manuscript confusing. I suggest separating them into two sections. In particular, more explanations of the preprocessing operations should be made, instead of only citing reference literature.

Great thanks for the comment. Based on the suggestion, the "Collection of global training samples" was split into an independent section 3. In addition, some arguments have been added more explanations

As the non-impervious surfaces consisted many land-cover types (water, vegetation, cropland and bare area) and some of them were spectrally similar to the impervious surface. For example, the bare soil and high reflectance impervious surfaces usually shared similar surface reflectance especially in arid and semi-arid areas with large areas of bare soils because the composition of impervious surfaces included rock material which was also found in bare areas (Sun et al., 2019b; Weng, 2012), the cropland showed similar reflectance to these low reflectance impervious surfaces (such as rural village, old cities) because they were usually composited of vegetation and high reflectance artificial materials or bare soils (Li et al., 2015). Therefore, the non-impervious training samples were split into three independent groups including: bare area, cropland and other non-impervious land-cover types. Furthermore, many studies had demonstrated that the distribution and balance of training samples had great influence on the mapping accuracy. For example, Zhu et al. (2016) found unbalanced training samples directly resulted in rare land-cover types under-represented relative to more abundant classes. Since the impervious surface was usually sparser than the non-impervious land-cover types (bare soil, cropland and so on), the training samples with uniform distribution were selected to ensure the rationality of training samples and capture all relevant spectral heterogeneity within impervious surfaces, namely, the approximate ratio of 1:3 was

used to represent the proportion of impervious to non-impervious surfaces (bare area, cropland and other non-impervious land-cover types).

Some parts of introduction to data preprocessing were also not understandable. Here is an example: "the suburban areas or rural villages were also easy to confused with croplands (Li et al., 2015)". It is not reasonable to compare a land use element (suburban area) with a land cover element (cropland). Explanations are always required for each of your arguments.

Great thanks for the comment. Yes, it was unreasonable to compare the land-use element with land-cover element, so this sentence has been rewritten as "the cropland showed similar reflectance to these low reflectance impervious surfaces (such as rural village, old cities) because they were usually composited of vegetation and high reflectance artificial materials or bare soils (Li et al., 2015)". The explanations of other arguments have been added according to the previous response.

Questions on accuracy assessment (Lines 315). Two accuracy assessment was conducted respectively in "fraction" way and "classified pixel" way. How much difference do the two accuracy assessment methods make? What special information can be provided by each method?

Great thanks for the comment. The detailed explanations of 'fraction-based validation' and 'sample-based validation' have been added as:

To completely analyze the performance of the MSMT_RF-based method, two validation methods including 'fraction-based' and 'pixel-based' were adopted. First, the 'fraction-based' validation method mainly illustrated the spatial agreement of impervious surfaces between the MSMT_RF-based impervious surface map and several existing products (GlobeLand30-2010, FROM_GLC-2015, NUACI-2015, HBASE-2010 and GHSL-2015) from a global perspective. Specifically, all these global 30-m impervious surface maps were aggregated to a resolution of $0.05\,°\times0.05\,°$ and the fraction of impervious area was then calculated. Following that, scatter plots of the linear regression between the MSMT_RF-based results and the reference data were produced to provide the quantitative metrics of the agreement, including coefficient of determination ($R^2$) and root mean square error (RMSE).

In addition, a 'pixel-based' validation method, based on the visual interpretation samples over fifteen $1\,°\times1\,°$ regions covering different impervious landscapes and continents, was used to quantitatively analyze the accuracy metrics, including overall accuracy (O.A.), producer's accuracy (P.A.), user's accuracy (U.A.) and kappa coefficient (Olofsson et al., 2014) for assessing the performance of the MSMT_RF-based global impervious surface mapping.

Questions on Figure 5. As mentioned in previous questions, further review of currently available global impervious surface maps is needed. In the revised manuscript, I suggest adding error bars for progressive fraction intervals (e.g. 0.05, 0.1, 0.15, 0.2, : : :, 1.0)

Great thanks for the comment. Based on the previous and this comment, we added two global impervious products (GHSL and HBASE). Except for this scatter plots, we have added the spatial variations of six global impervious products as:

[Figure]

**Figure 7. The spatial variations of six global 30-m impervious products after aggregating to the resolution of 0.05°.**

Based on the suggestion, the error bars for the progressive intervals of 0.05 have been added as:

[Figure]

**Figure 8: Scatter plots between the MSMT_RF-based impervious map and the GlobeLand30-2010, FROM_GLC-2015, NUACI-2015, GHSL-2015 and HBASE-2010 global impervious surface products at a spatial grid of 0.05°×0.05°.**

Confusing parts in Section 4.2. "As the stratified random sampling strategy was applied to each validation region independently, the low and medium density regions were easier to select these mixed impervious validation points (simultaneously containing the impervious and non-impervious surfaces in the 30-m validation window and the impervious areas exceed the predefined threshold of 50%) which were most difficult to identify for impervious surface mapping (Line 380)." What information did you want.

Great thanks for the comment. The meaning of this sentence was that the impervious surface mapping usually suffered relatively low accuracy over low and medium density regions where contained a higher proportions of mixed impervious surfaces (simultaneously containing the impervious and non-impervious surfaces in the Landsat pixel and the impervious areas exceed the

threshold of 50%). In the revised manuscript, these confusing sentences have been removed.

Questions for Figure 6. It is clear to show difference between impervious surface maps but not clear to visually compare RGB pixels with your maps. The RGB satellite images of macro areas may not be suitable to compare it with classified land cover map. Subset urban areas are preferred so that readers can clearly see how well the map is classified. Besides, I may not agree that "low, medium, high- density" areas are representative for comparison. To improve the figure, I suggest globally selecting urban areas with different landscapes (e.g. desert landscape urban areas such as Phoenix city, vegetation prevalent cities such as New York City). Furthermore, please do more works in reviewing global impervious surface datasets.

Great thanks for the comment. Based on the suggestion, this figure has been expanded as:

[Figure]

[Figure]

Figure 10: Comparisons between the MSMT_RF-based maps and other impervious surface products (corresponded to the NUACI products developed by Liu et al. (2018), the FROM_GLC products developed by Gong et al. (2013), the GHSL products developed by Florczyk et al. (2019), the GlobeLand30 products developed by Chen et al. (2015), and the HBASE products developed by Wang et al. (2017a), respectively) for five regions with various impervious landscapes.

Figure 7 is an experiment result, and it should be moved to "Result"
Great thanks for the comment. The section has been moved to the "Result" as the Section 5.1 'The importance of multi-source and multi-temporal features'.

"The importance of all 37 training features for the six regions is illustrated in Fig. 7. These results indicate that the Sentinel-1 SAR features (VV and VH) had the greatest contribution to the final decision in most regions because SAR images can provide information about the structure and dielectric properties of the surface materials (Line 440)". What VV and VH feature difference is revealed between different land covers (e.g. impervious surface, forest, croplands, bare soil, water)?
Great thanks for the comment. Based on the suggestion, the response of different land-cover types over optical and SAR imagery have been added as:
To intuitively understand the characteristics of different land-cover types on optical and SAR imagery, two regions (the vegetation-prevalent region of Asia and bare soil-prevalent semi-arid region of Australia) were selected for comparison analysis. Fig. 4 illustrated the reflectance and backscatter statistics (mean and standard deviation) of five typical land-cover types (cropland, vegetation, bare soil,

impervious surfaces and water body). Obviously, impervious surfaces had highest backscatter signals in VV because of the high dielectric properties of the building materials, the unique geometry of manmade features, and the special radar echo properties of artificial structures, followed by the vegetation land-cover types. Further, since only a small part of the polarized signals (vertical turning horizontal) were returned to the sensor, the VH was significantly lower than VV but the ranking orders of different land-cover types in VH was similar to that of VV. Due to the complicated construction and heterogeneity of the impervious surfaces, the impervious surfaces also had highest standard deviation, for example, the urban central usually reflected higher VV and VH signals than the village buildings. If only Sentinel-1 SAR features were used to identify impervious surfaces, there would be serious confusion between the mountainous vegetation with low reflectance impervious surfaces (such as: villages and small cities), fortunately, the optical reflectance features performed well to distinguish them because of significant spectral differences. However, if only the multi-temporal optical imagery were used to detect the impervious surfaces, there would be obvious confusion between impervious surfaces with bare soils and croplands, for example, the spectral characteristics of impervious surfaces, bare soils and croplands were overlapping in the Asia region (Fig. 4). In summary, only the combination of multi-source training features could guarantee the classification accuracy across different impervious landscapes.

[Figure]

**Figure 4: The reflectance/backscatter characteristics of different land-cover types over Landsat optical and Sentinel-1 SAR imagery in the Asia and Australia regions.**

"Similarly, Zhu et al. (2012) demonstrated that the inclusion of multi-temporal imagery increased the accuracy by 8.9%. Schug et al. (2018) also found that bi-seasonal information could produce a more reliable performance than a single-year composited image. Therefore, temporal variability can be considered an important addition to accurate impervious surface mapping (Line 455)." Discussion and explanation should be made. Please exactly explain the theory in which how these datasets work for improving classification accuracies. Which land cover accuracy is improved by including these datasets?

Great thanks for the comment. The reasons why the temporal variability was important for impervious mapping have been added as:

The reasons that the temporal information was important for accurately mapping of impervious

surface included: (1) some land-cover types such as cropland had similar spectra with impervious surface at fallow season, but with the growing season imagery imported, this misclassification could be easily removed; (2) Sun et al. (2017) explained that the growing season was the best time for impervious surface mapping over temperate continental climate zones, and Zhang et al. (2014a) found that winter (dry season) is the best season to estimate impervious surface in subtropical monsoon regions. The multi-temporal information can address the problem of seasonal variability at different geographical zones. Fig. 4 (Australia region) also illustrated that the cropland and impervious surfaces were spectrally inseparable in the 15th percentile but the difference was obvious in the 85th percentile. Therefore, temporal variability can be considered an important contribution for accurate impervious surface mapping.

"Similarly, Clarke et al. (1997) explained that topographical variables (slope, aspect and DEM) contribute a lot to impervious surface mapping. These features are, therefore, indispensable in the accurate mapping of impervious surfaces in complex landscapes (Line 465)." Clarke et al. (1997) was cited without further explanation. Readers would like to know mechanism of topographical variable contributing to impervious surface mapping?

Great thanks for the comment. The mechanism of topographical variable contributing to impervious surface mapping has been added as:

Lastly, since most regions are located in the flat areas, only the cumulative importance of topographical variables over the region in Asia exceeded 5%. The reasons why topographical information reached high importance over mountainous areas were because the impervious surfaces usually located in the flat areas (Ban et al., 2015) and Sentinel-1 SAR imagery had high backscatter signals over mountainous areas similar to the impervious surfaces, which increased the importance of topographical variables. Similarly, Clarke et al. (1997) explained that topographical variables (slope, aspect and DEM) contribute a lot to impervious surface mapping over mountainous areas. These features are, therefore, indispensable in the accurate mapping of impervious surfaces in complex landscapes

---

## Author Comment (AC2) · 2 May 2020

**Response to comments**

**Paper #:** essd-2019-200
**Title:** Development of a global 30-m impervious surface map using multi-source and multi-temporal remote sensing datasets with the Google Earth Engine platform
**Journal**: Earth System Science Data

**Reviewer #2**

The paper describes a new impervious surface dataset developed by combining several remote sensing instrumentation at 30m resolution. As described in the introduction, several datasets describing impervious studies exist at a global scale. The strength of this paper is in my opinion the use of multi-sensor information and the use of an open-source platform the generate these maps (Google Earth Engine). Furthermore, a relatively good accuracy of the map is achieved compared to three other impervious surface products. The paper is very well written and is easy to follow. The introduction also gives a very good overview of current existing literature. The paper is very mature and contains all information one would expect for this kind of work. Most of the comments that popped in my mind while reading the paper were assessed later in the manuscript. As such, for me only minor revisions are necessary. I describe some comments below.

Great thanks for the comment. The manuscript has been improved according to your and other reviewers' comments.

General comments
- Training points are achieved from Globeland30 and are not independent based on independent experts (which is done for the validation data). Several checks are done on the training data, but you are still using a derived product with errors to train your model. In the discussion, this problem is assessed (section 5.2). However, I would state this more clear that the training sample can contain errors in the material and methods section and potentially move the discussion to the material and methods section or refer in the material and methods section that this problem will be assessed later.

Great thanks for the comment. According to the suggestion, we added a paragraph to explain that the training sample may contain some error, because they were collected from Globeland30. The detailed explanations are listed in the Method Section "3 Collection of global training samples" as: "Although a series of rules were applied to guarantee the high confidence of global training samples, due to the classification error in GlobeLand30 and the temporal interval between GlobeLand30 and input imagery, the global training dataset inevitably contained some erroneous samples. The relationship between the percentage of the erroneous samples and the mapping accuracy of impervious surface was analyzed in the Discussion section 6.1, and the results indicated that the error in the training samples had little effect on the mapping accuracy."

- Only homogeneous training points from Globeland30 are included. Therefore, the training points are always clear impervious surfaces leading to only clear impervious surfaces to be classified later. Don't you underestimate the total amount of impervious surfaces then in your final product? How does the total % of impervious surface compare to Globeland30, GLC and NAUCI for the

globe? This can maybe be compared to the results presented in figure 5.

Great thanks for the comment. Although only homogeneous training points from GlobeLand30 are included, the accuracy assessment in the section 5.4 has demonstrated that the proposed method achieved lower omission error than other products (NUACI-2015, FROM_GLC-2015, GHSL-2015, GlobeLand30-2010 and HBASE-2010).

From the perspective of the value of the user's accuracy for impervious surfaces, the MSMT_RF method performed better than the other impervious surface products (meaning lower omission error) achieving the accuracy of 0.932, especially in the cropland-prevalent and vegetation-prevalent impervious landscapes (such as: Bangkok, Winnipeg, Xi'an…). Specifically, NUACI-2015 had the lowest user's accuracy of 0.562 and this might be due to its poor performance over small impervious surfaces (Sun et al., 2019b). FROM_GLC-2015 had a similar performance with the MSMT_RF method for big cities (such as New York, Moscow and Johannesburg), but its accuracy decreased sharply over 'small-city' regions (such as Lhasa, Winnipeg). The performance of GHSL-2015 was closest to the MSMT-2015 over most validation regions, but it also missed the fragmented objects (villages and roads) over cropland-prevalent city (such as Bangkok and Winnipeg). As the minimum mapping unit of GlobeLand30 was a 4×4-pixel area, many rural impervious surfaces were ignored in these validation regions, which caused large omission errors of 23.9%. Finally, partly due to the 5 years' interval between the HBASE-2010 and validation samples, HBASE-2010 also suffered the omission error of 12.5%.

In addition, we analyzed the spatial variations of six global impervious products at the spatial resolution of 0.05°, the figure also indicated the proposed method gave consistent mapping with other datasets.

[Figure]

Figure 7: The spatial patterns of six global 30-m impervious products after aggregating to the resolution of 0.05°.

Finally, the total impervious areas of NUACI-2015, FROM_GLC-2015, GHSL-2015, GlobeLand30-2010 and HBASE-2010 were 49.53%, 54.67%, 78.55%, 67.76% and 97.24% that of the MSMT-2015 (our study), respectively.

- The validation points are retrieved from 12 regions. How representative are these regions for the globe? Since impervious areas might have very different characteristics depending on the region. For Africa for example, the validation points are achieved for two big cities only.

Great thanks for the comment. Yes, we agree that impervious areas might have very different characteristics depending on the region. However, it was a time-consuming task to collect validation samples over the globe. According to the comment and suggestion from your and other

reviewers, to make the validation regions more representative, we re-selected these regions by combining the impervious landscapes, for example, desert landscape urban areas such as Phoenix city, vegetation prevalent cities such as New York City. Specifically, the section 2.3 "validation samples" was changed as:

"To quantitatively assess the performance of the global impervious surface datasets, fifteen validation regions, covering different continents and various urban landscapes (the bare soil prevalent cities: Phoenix (PNX), Madrid (MDR), Riyadh (RYH), Niamey (NIM), Johannesburg (JHB), Ntuman (NTU) and Lhasa (LHS), vegetation prevalent cities: New York (NYK), Manaus (MNS), Moscow (MSC), San Paulo (SPL) and Melbourne (MBN), as well as cropland prevalent cities: Winnipeg (WIP), Bangkok (BGK) and Xi'an (XAN)), were selected (Fig. 1). For each validation region, 600‑1000 samples were randomly generated using the stratified random sampling strategy (Bai et al., 2015)."

[Figure]

Figure 1: The spatial distribution of the fifteen validation regions (blue) corresponding to regions of different impervious landscapes on different continents, together with the six 5°×5° validation regions (red) used to measure the variable importance.

Specific comments

- Globeland30 data from 2010 is used as training data. How do you account for changes in urban areas between 2010-2015? You state that there is an irreversible state from non-impervious to impervious surfaces, but this could mean that some non-impervious surfaces in 2010 have now changed to impervious in 2015.

Great thanks for the comment. As there was temporal interval of 5years between GlobaLand30 and input imagery, we assumed that the process of transforming non-impervious surfaces into impervious surfaces was irreversible during the period 2010 to 2015, meaning that the global impervious training samples derived from GlobeLand30-2010 could also be used to represent the situation in 2015.

However, it was possible that the non-impervious pixels in 2010 were transformed into impervious surfaces in 2015. Therefore, some non-impervious training samples in Globeland30-2010 would be impervious surface in 2015. In order to mitigate the problem, the EANTLI light data was used to remove these changed samples. For example, the non-impervious training samples with high EANTIL value ($EANTLI = \frac{1+(NTL_{norm}-EVI)}{1-(NTL_{norm}-EVI)} \times NTL$) in 2015 would be removed. Detailed explanation was revised in the section of "3. Collection of global training

samples" as:

"As for the non-impervious pixels, there was usually a negative correlation between non-impervious surfaces and EANTLI values, and the non-impervious surface samples turned into impervious surface would have high EANTLI values in 2015, so if the cumulative probability of a candidate non-impervious point in CanTPS_Imp was greater than the top 20th percentile of the cumulative probability of all candidate non-impervious points (the threshold being based on the overall accuracy of 80.33% for GlobeLand30-2010 and a little potential conversion samples), the candidate non-impervious point was also removed."

- Add to table 1 that the 15 + 85 percentile are used for the Landsat bands and vegetation indices.

Great thanks for the comment. It has been added as:

Table 1. Training features for global impervious surface mapping.

| Data | Features | References |
|---|---|---|
| LandSat-8 OLI | Reflectance: 15th and 85th percentiles of Blue, Green, Red, NIR, SWIR1 and SWIR2 | Liu et al. (2018) |
| | Normalized indices: 15th and 85th percentiles of NDVI, NDWI and NDBI | |
| | Textural variables: variance, dissimilarity and entropy of the NIR | Chen et al. (2016) |
| Sentinel-1 SAR | Annual statistics: mean and standard deviation of VV and VH | Sun et al. (2019b) |
| | Textural features: dissimilarity, variance and entropy of VV and VH | Zhang et al. (2014b) |
| DEM | Elevation, slope and aspect | Clarke et al. (1997) |

- Line 265, remove 'the'

Great thanks for the comment. It has been removed as:

"In addition, as Sun et al. (2017) explained that the growing season was the best time for impervious surface mapping over temperate continental climate zones..."

---

## Author Comment (AC3) · 2 May 2020

**Response to comments**

**Paper #:** essd-2019-200
**Title:** Development of a global 30-m impervious surface map using multi-source and multi-temporal remote sensing datasets with the Google Earth Engine platform
**Journal**: Earth System Science Data

**Reviewer #3**

This paper presents a new global 30m impervious surface map produced with multi-source and multi-temporal remote sensing datasets and random forest (MSMT_RF). Compared with the currently available impervious products (i.e., GlobeLand30, FROM_GLC and NUACI), this MSMT_RF-based product has higher overall accuracy and kappa coefficient, which are 96.6% and 0.90, respectively. The superiority of the MSMT_RF-based product stems from two significant innovations of the method proposed in this study. First, multi-source and multi-temporal remote sensing data are combined to produce the impervious surface map. The comprehensive information provided by the combined data is useful in classifying land cover types, so the superiority of the MSMT_RF-based product in comparison with the other products is convincing. Second, a novelty method is proposed for selecting training samples based on the available impervious product and VIIRS NTL and MODIS EVI imagery. This method allows for the fully automatic selection of training samples to avoid manual training sample selection, which is time-consuming and laborious, especially at a global scale. This method has significant implications for producing more perfect global data products based on existing data products. I believe this study is a breakthrough over previous works in impervious surface mapping and will appeal to a broad readership. However, there are still some minor issues that should be addressed before final publication.
Great thanks for the positive comment. The manuscript has been improved according to your and other reviewers' comment.

Line 35, "urban the environment" should be "urban environment"
Great thanks for the comment. It has been corrected.

Figure 1, I cannot see the blue rectangles but only black points, which are supposed to be the blue rectangles. The authors should figure out how to make blue rectangles clear.
Great thanks for the comment. As we re-selected the validation regions based on the impervious landscapes, the new spatial distribution figure was changed as:

[Figure]

Figure 1: The spatial distribution of the fifteen validation regions (blue) corresponding to regions of different impervious landscapes on different continents together with the six 5°×5° validation regions (red) used to measure the variable importance.

Why did the authors select training samples based on Globe30 product but not FORM_GLC, which is also a 2015 product and seems to be more appropriate? Please elaborate.

Great thanks for the comment. The reasons why we chose the GlobeLand30 instead of FROM_GLC have been added as:

"The GlobeLand30 land-cover product was used to derive global training samples because it had many advantages including: (1) the impervious surface layer in GlobeLand30 was accurately developed by combining the pixel-based classification, multi-scale segmentation and manual editing based on high resolution imagery and validated to achieve an user's accuracy of 86.7%; (2) it simultaneously contained the impervious surface and other land-cover types similar to impervious surface (such as cropland and bare land), so the global training samples including several non-impervious land-cover types could be easily collected to build the RF model for accurately mapping of impervious surface."

Figure 5, please provide the label of axes.

Great thanks for the comment. The label of axes were added as:

[Figure]

**Figure 8: Scatter plots between the MSMT_RF-based impervious map and the GlobeLand30-2010, FROM_GLC-2015, NUACI-2015, GHSL-2015 and HBASE-2010 global impervious surface products at a spatial grid of 0.05°×0.05°. The error bars were the standard deviation between reference datasets with fitted results.**

Table 2. How the different categories, e.g., high, low, medium, are defined? Are they defined quantitatively or subjectively? Please elaborate.

Great thanks for the comment. The impervious surface density (low, medium and high) was defined by combining the histogram of impervious areas at 0.05°×0.05°. In the revised manuscript, we re-selected the validation regions through the land-cover landscapes according to the suggestion of Reviewer 1. Specifically:

"To quantitatively assess the performance of the global impervious surface map, fifteen validation regions, covering different continents and various urban landscapes (the bare soil prevalent cities: Phoenix (PNX), Madrid (MDR), Riyadh (RYH), Niamey (NIM), Johannesburg (JHB), Ntuman (NTU) and Lhasa (LHS), vegetation prevalent cities: New York (NYK), Manaus (MNS), Moscow (MSC), San Paulo (SPL) and Melbourne (MBN), as well as cropland prevalent cities: Winnipeg (WIP), Bangkok (BGK) and Xi'an (XAN)), were selected."

Figure 6. I suggest the authors to provide the location information (e.g., city name or latitude-longitude grid) of these areas. It will allow readers to check ground truth in Google Earth.

Great thanks for the comment. The city names were added as:

[Figure]

[Figure]

**Figure 9: Comparisons between the MSMT_RF-based maps and other impervious surface products (corresponded to the NUACI products developed by Liu et al. (2018), the FROM_GLC products developed by Gong et al. (2013), the GHSL products developed by Florczyk et al. (2019), the GlobeLand30 products developed by Chen et al. (2015), and the HBASE products developed by Wang et al. (2017a), respectively) for five regions with various impervious landscapes.**

Page 20, the authors found that the importance of Landsat textural features is low, whereas previous studies confirmed the contribution of textural features to impervious mapping. More explanations can be given on this contradiction. One possible explanation may be the different data sets. Many studies have indicated that textural information is helpful in land cover classification, especially in high-resolution images. Shaban and Dikshit (2001) used the textural information in SPOT images, while the authors used that in Landsat-8 images. The difference in spatial resolution may cause the different contribution of textural features in impervious surface mapping.

Great thanks for the comment. Actually, as the SAR backscatter and texture features also had ability to provide information on the structure and variability properties of surface materials, the importance of Landsat textural features was low. If only considering the optical Landsat imagery, the importance of Landsat textual features were significantly improved. This reasons have been added as:

"Thirdly, the importance of Landsat texture features was lower than 5% in these six regions because the Sentinel-1 SAR backscatter and texture features were able to provide information on the surface material and its spatial structure and variation. Due to the complexity of land-surfaces and different mechanism of optical and SAR imagery, the optical textures could complement a lot to SAR features at mountainous and semiarid areas (Asia and Australia regions). Some studies demonstrated that these features contributed a lot to the improvement of impervious mapping accuracy. For example, Shaban and Dikshit (2001) emphasized that the integration of texture variables increased the accuracy from 86.86% to 92.69% because texture imagery could capture the local spatial structure and the variability of land-cover categories."

Page 20, I agree with that the improvement made by this study is mainly due to the combination of the multi-source and multi-temporal information, but it may be misleading to state that the classification-based method performed better than spectral index-based method since they are performed based on the different data sets. I do not think the classification-based method can

achieve a high accuracy only with Landsat data.

Great thanks for the comment. Yes, it was misleading to state that classification-based method performed better than the spectral index-based method, the improvement of mapping accuracy was mainly due to the combination of the multi-source and multi-temporal information. Therefore, we removed these misleading paragraph in the revised manuscript.

---

## Author Response (AR2)

Dear Topical Editor and Reviewers:

On behalf of my co-authors, we thank you very much for reviewing our manuscript and giving us the opportunity to revise the manuscript. We appreciate the comments on our manuscript entitled "Development of a global 30-m impervious surface map using multi-source and multi-temporal remote sensing datasets with the Google Earth Engine platform" (essd-2019-200). We have revised the manuscript carefully according to the comments. All the changes were high-lighted (red and green color) in the manuscript. And the point-by-point response to the comments of the reviewers is also listed below.

Looking forward to hearing from you soon.

Best regards,

Prof. Liangyun Liu
liuly@radi.ac.cn
Institute of Remote Sensing and Digital Earth, Chinese Academy of Sciences
No.9 Dengzhuang South Road, Haidian District, Beijing 100094, China

This paper generates an accurate global coverage impervious surface map with multiple remote sensing datasets. The generated map was proven to have optimal accuracy measurement compared with other state-of-art impervious surface products. I appreciate the author's response to my questions in the last round of review. I agree that this paper can be published after a minor revision.

Great thanks for the positive comment. The manuscript has been further improved according to your comments.

1. Line 20-25. "First, the global impervious and non-impervious training samples were automatically derived by combining the GlobeLand30 land-cover product with VIIRS NTL and MODIS enhanced vegetation index (EVI) imagery. Then, based on global training samples and multi-source and multi-temporal imagery, a random forest classifier was trained and used to generate corresponding impervious surface maps for each 5 °×5 ° cell of a geographical grid. Finally, a global impervious surface map, produced by mosaicking numerous 5 °×5 ° regional maps, was validated by interpretation samples and then compared with five existing impervious products (GlobeLand30, FROM_GLC, NUACI, HBASE and GHSL). "

At the beginning of abstract readers are informed this work will present a global impervious surface map at 30 m resolution, but why did the authors introduce the map was generated for each 5 °×5 ° geographical grid? please double-check the content.

Great thanks for the comment. Actually, there were two kinds of models used for generation of a global impervious surface product - global modeling (building a single classifier using global training data) and local adaptive modeling (dividing the globe into a number of geographical grids and then building local classifiers using corresponding training samples at each grid). Many studies have demonstrated that the local adaptive model performed better than the single global classification model. Therefore, we first split the global land surface into 954 5 °×5 ° geographical tiles and then trained local adaptive classifiers within each geographical tile.

The **Abstract** has been revised as:

Then, the local adaptive random forest classifiers, allowing a regional adjustment of the classification parameters to take into account the regional characteristics, were trained and used to generate regional impervious surface maps for each 5 °×5 ° geographical grid using local training samples and multi-source and multi-temporal imagery. Finally, a global impervious surface map, produced by mosaicking numerous 5 °×5 ° regional maps, was validated by interpretation samples and then compared with five existing impervious products (GlobeLand30, FROM_GLC, NUACI, HBASE and GHSL).

The **Section** 4.2 "Random forest classification model" has been revised as:

There were two kinds of models used for generating a global impervious surface product – global modeling (building a single classifier using global training data) and local adaptive modeling (dividing the globe into a number of regions and then building local classifiers using corresponding regional training data). For example, Gong et al. (2013) built a single global classifier using 91,433 training samples to produce the FROM_GLC land-cover products; Bontemps et al. (2010) first split the world into 22 ecological regions, and then trained the classifier for each region using local training samples to produce the GlobeCover2009 land-cover products. Recently, Zhang and Roy (2017) demonstrated that the local adaptive model performed better than the single global classification model; While Radoux et al. (2014) found that using a local window increased the

sensitivity to the quality of the training dataset. Therefore, after balancing the data volume, computation efficiency and classification accuracy, we first split the global land surface into 954 5 °×5 °geographical tiles, and then trained local adaptive classifiers for each geographical tile. In addition, to ensure the classification consistency across neighboring geographical tiles, the training data from adjacent 3 ×3 tiles were imported to train the random forest classifier and classify the central tile.

2. Line 59. "GlobeLand30 combined the pixel-based classification, segmentation and manual editing based on the high resolution imagery to develop the 30-m impervious surface map as an independent layer with the user's accuracy of 0.867 (Chen et al., 2015)."

GlobeLand30 provides users with an "artificial surface layer" instead of an "impervious surface map". In the original literature which released Globeland30, the "artificial layer" was defined as follows: "Artificial surfaces mainly consist of urban areas, roads, rural cottages and mines, which are primarily based on asphalt, concrete, sand and stone, bricks, glasses, and other materials. They can be divided into three subclasses including high reflectance, low reflectance and vegetated type." (Chen, J., Chen, J., Liao, A., Cao, X., Chen, L., Chen, X., ... & Zhang, W. (2015). Global land cover mapping at 30 m resolution: A POK-based operational approach. ISPRS Journal of Photogrammetry and Remote Sensing, 103, 7-27.)

I agree that the major content of artificial surfaces is represented by impervious surfaces, but other small vegetation patch areas should not be simply ignored. I suggest the authors should make more clarifications towards this issue, instead of directly using "artificial surfaces" as "impervious surfaces". Discussion about the impact of "artificial surfaces" can be made in your paper.

Great thanks for the comment. The artificial surfaces in the GlobeLand30-2010 were detailedly described in the work of (Chen et.al 2016, **Global mapping of artificial surfaces at 30-m resolution**), the vegetated artificial surfaces were more similar to the definition "**green artificial surface that shows much vegetation spectral signals, such as rural residences and residential zones with good greenness.**"

Chen, X., Cao, X., Liao, A., Chen, L., Peng, S., Lu, M., Chen, J., Zhang, W., Zhang, H., and Han, G.: Global mapping of artificial surfaces at 30-m resolution, Science China Earth Sciences, 59, 2295-2306, https://doi.org/10.1007/s11430-016-5291-y, 2016.

Based on the suggestion, these small vegetation patches in the "artificial surfaces" have been explained in the Section 3 as:

It should be noted that the definition of artificial surfaces in the GlobeLand30 was slightly different from the impervious surfaces in this study. Specifically, artificial surfaces in the GlobeLand30 were divided into three subclasses including high reflectance, low reflectance and vegetated type (Chen et al., 2015), and a small part of purely vegetated artificial surfaces (such as small vegetation patches in the residential zones with good greenness) actually didn't belong to the impervious surfaces. Fortunately, the ENATLI, measuring the likelihood of the pixel corresponding to an impervious surface, usually revealed the low values over these vegetation patches. Therefore, these purely vegetated artificial surface pixels could be removed from the

CanTPS_Imp using the lowest 15th quantile of the cumulative probability of all candidate impervious points for EANTLI-2015.

3. Line 158. "The Shuttle Radar Topography Mission digital elevation model (SRTM DEM), provided by the NASA JPL at a resolution of 1 arc-second (approximately 30 m) and covering the area between 60 ° north and 56 ° south (Farr et al., 2007), was a useful auxiliary dataset for impervious surface mapping over mountainous areas because impervious surfaces mainly located in the flat areas and Sentinel-1 SAR data usually reflected high backscatter similar to the impervious surfaces in mountainous areas."

The reason for using the DEM dataset for classification is not clear. There are a lot of cities being built on mountains or having hills, e.g. Lisboa, Porto, Prague, Rome, et al. The DEM dataset may cause confusion in classification. Such negative influences may need to be discussed in your manuscript.

Great thanks for the comment. It should be noted that the topographical variables (elevation, slope) were imported as the classification features to train the local random forest classifier instead of the classification rules (for example, Sun et al. (2019) used a slope threshold of 15 ° to exclude impervious surfaces over steep region).

Sun, Z., Xu, R., Du, W., Wang, L., and Lu, D.: High-Resolution Urban Land Mapping in China from Sentinel 1A/2 Imagery Based on Google Earth Engine, Remote Sensing, 11, 752, https://doi.org/10.3390/rs11070752, 2019b.

In this study, if the impervious surfaces and non-impervious surfaces simultaneously located in the mountain areas, the importance of topographical variables would decrease in the local random forest classifier. For example, Figure 3 illustrates the importance of all 37 training features for the six regions. The second region (Europe in Figure 1), located on the Mediterranean coast and contained complex mountain areas, had low importance for three topographical variables.

[Figure]

Figure 1: The spatial distribution of the fifteen validation regions (blue) corresponding to regions of different impervious landscapes on different continents together with the six 5 °×5 ° validation regions (red) used to measure the variable importance.

[Figure]

Figure 3: The importance of the input features derived from the random forest model using the training samples in six continental regions.

Figure 1s also illustrated the performance of our impervious maps over four mountainous cities (Lisboa, Porto, Prague and Rome), the results showed that the impervious surfaces over terrains were correctly identified in the mountain cities.

[Figure]

Figure 1s. The performance of our impervious maps over four mountainous cities (Lisboa, Porto, Prague and Rome).

Specifically, the reason for using the DEM dataset for classification has been revised as:

[revised manuscript text omitted]

---

## Author Response (AR3)

Dear Topical Editor:

On behalf of my co-authors, we appreciate the comments on our manuscript entitled "Development of a global 30-m impervious surface map using multi-source and multi-temporal remote sensing datasets with the Google Earth Engine platform" (essd-2019-200).

We have revised the manuscript carefully according to the comments. All the changes were high-lighted (red and green color) in the manuscript. And the point-by-point response to the comments of the reviewers is also listed below.

Looking forward to hearing from you soon.

Best regards,

Prof. Liangyun Liu

liuly@radi.ac.cn

Institute of Remote Sensing and Digital Earth, Chinese Academy of Sciences

No.9 Dengzhuang South Road, Haidian District, Beijing 100094, China

1) Many acronyms used but not systematically defined. E.g. you defined your multi-source multi-temporal random forest (MSMT_RF) acronym at lines 314, 315 but you have already used it on line 184. Please assure all acronyms receive proper initial definition. Perhaps you need a list of acronyms as an appendix?

Thanks for the comment. The acronyms have been systematically revised in the manuscript, and the acronyms list has been added in the appendix as:

**Table A1 List of abbreviations and acronyms**

| Abbreviation | Definition |
| --- | --- |
| DEM | Digital elevation model |
| DMSP OLS | Defense Meteorological Satellite Program Operational Linescan System |
| EANTLI | EVI-adjusted night-time light index |
| EVI | Enhanced vegetation index |
| FROM_GLC | Finer Resolution Observation and Monitoring of Global Land Cover |
| GEE | Google Earth Engine |
| GHSL | Global 30-m Human Settlement Layer |
| GlobeLand30 | 30 m Global Land Cover data product |
| HBASE | Human Built-up and Settlement Extent |
| MSMT_RF | Multi-source, multi-temporal random forest classification |
| NDBI | Normalized Difference Built-up Index |
| NDVI | Normalized Difference Vegetation Index |
| NDWI | Normalized Difference Water Index |
| NLCD | National Land Cover Dataset |
| NTL | Night-time light |
| NUACI | Normalized Urban Areas Composite Index |
| $R^2$ | Coefficient of determination |
| RMSE | Root mean square error |
| RF | Random forest |
| SAR | Synthetic aperture radar |
| VIIRS | Visible Infrared Imaging Radiometer |

2) You use kappa coefficient as a defining statistical indicator of quality / veracity, starting even in the abstract. As researchers familiar with feature identification in satellite images, you have a preferred definition and understanding of kappa coefficient. Most readers will not share your experience. Please define kappa coefficient as you apply it here. It needs a citation?

Thanks for the comment. The definition of kappa coefficient has been added in the Abstract as:

[revised manuscript text omitted]